# The minimal meningococcal ProQ protein has an intrinsic capacity for structure-based global RNA recognition

Saskia Bauriedl[1,5], Milan Gerovac [2,5], Nadja Heidrich[2], Thorsten Bischler [2], Lars Barquist [3,4], Jörg Vogel [2,3,6 ✉] & Christoph Schoen [1,6 ✉]

FinO-domain proteins are a widespread family of bacterial RNA-binding proteins with regulatory functions. Their target spectrum ranges from a single RNA pair, in the case of plasmid-encoded FinO, to global RNA regulons, as with enterobacterial ProQ. To assess whether the FinO domain itself is intrinsically selective or promiscuous, we determine in vivo targets of *Neisseria meningitidis*, which consists of solely a FinO domain. UV-CLIP-seq identifies associations with 16 small non-coding sRNAs and 166 mRNAs. Meningococcal ProQ predominantly binds to highly structured regions and generally acts to stabilize its RNA targets. Loss of ProQ alters transcript levels of >250 genes, demonstrating that this minimal ProQ protein impacts gene expression globally. Phenotypic analyses indicate that ProQ promotes oxidative stress resistance and DNA damage repair. We conclude that FinO domain proteins recognize some abundant type of RNA shape and evolve RNA binding selectivity through acquisition of additional regions that constrain target recognition.

---

[1] Institute for Hygiene and Microbiology, University of Würzburg, 97080 Würzburg, Germany. [2] Institute for Molecular Infection Biology (IMIB), University of Würzburg, 97080 Würzburg, Germany. [3] Helmholtz Institute for RNA-based Infection Research (HIRI), Helmholtz Centre for Infection Research (HZI), 97080 Würzburg, Germany. [4] Faculty of Medicine, University of Würzburg, 97080 Würzburg, Germany. [5] These authors contributed equally: Saskia Bauriedl, Milan Gerovac. [6] These authors jointly supervised this work: Jörg Vogel, Christoph Schoen. ✉email: joerg.vogel@uni-wuerzburg.de; cschoen@hygiene.uni-wuerzburg.de

lobal post-transcriptional regulatory networks involving a central RNA-binding protein (RBP) and hundreds of mRNAs and small regulatory RNAs (sRNAs) have long been known in bacteria, centered on two RBP families, Csr/Rsm, and Hfq[1]. The recent discovery of ProQ as another global RPB in *Escherichia coli* and *Salmonella enterica*[2] demonstrated that additional such networks exist, at least in γ-proteobacteria. ProQ was found to target a distinct set of sRNAs and mRNAs rivaling the target regulons of CsrA or Hfq in size[3,4]. ProQ activity affects diverse pathways including motility and pathogenesis[2], and in *Salmonella* this RBP is required for full virulence[5].

Many important aspects of ProQ biology remain poorly understood, including the function of most associated sRNAs. Given the protein's documented RNA chaperone activity[6], these sRNAs may regulate mRNAs by base pairing mechanisms, as recently demonstrated for ProQ-dependent repression of the synthesis of histone-like protein HU-α[7]. Also unknown is how ProQ recognizes its targets with specificity. UV-crosslinking coupled with RNA-seq (UV CLIP-seq) in vivo has detected hundreds of ProQ sites in cellular transcripts but failed to reveal a common sequence motif[4]. In fact, this RBP seems novel in that it may govern a global post-transcriptional network by recognizing RNA structure or shape rather than primary sequence.

ProQ seems an unlikely candidate for a global RBP, as it belongs to the PF04352 Pfam family of FinO-domain proteins whose other two characterized members—FinO and RocC—perform specialized functions with a small number of RNA targets[8–10]. The plasmid-encoded FinO protein modulates conjugation in *E. coli* via a single sRNA–mRNA pair, while RocC controls natural transformation in *Legionella pneumophila* through one sRNA and four mRNAs of competence genes[11]. This unusual variation in RNA target number for the same single RNA-binding domain (RBD) raises the fundamental question of whether high selectivity (FinO, RocC) or global activity (ProQ) represents the major mode-of-action in this emerging RBP family. To address this, we here set out to determine the in vivo RNA target suite and physiological roles of one of the smallest family members known (Fig. 1a), the ProQ protein of *Neisseria meningitidis*.

The meningococcal ProQ (a.k.a. NMB1681 in *N. meninigitidis* strain MC58) represents an almost pure FinO domain, lacking the amino or carboxy terminal extensions carried by other family members[8]. Its crystal structure suggests a strong similarity to *E. coli* ProQ[8,12], and in vitro assays with artificial RNA substrates have demonstrated RNA binding, strand-exchange and duplexing activities[12]. However, in contrast to the recently established role of Hfq as an important RBP[13,14] in *N. meningitidis*, neither the physiological role nor possible in vivo RNA targets of ProQ have been investigated in any β-proteobacterium.

Here, we use in vivo UV CLIP-seq and global gene expression profiling to draft a ProQ-RNA interaction landscape and analyse its functional overlap with the Hfq targetome in the meningococcus. We find that ProQ interacts with >7% of all transcripts, both coding and noncoding, primarily at Rho-independent terminators. Our data indicate that ProQ stabilizes associated sRNAs and partially acts in concert with Hfq. By phenotypic screening of a *proQ* deletion mutant we identify a role for ProQ in DNA damage repair after UV light irradiation and oxidative stress responses. Together, these results establish ProQ as a second global RBP in *N. meninigitidis*, an organism that possesses Hfq but lacks a translational repressor of the near-ubiquitous Csr/Rsm family. Importantly, our findings also suggest conclusions as to the functional evolution of FinO domain proteins.

## Results

**General characteristics of the meningococcal ProQ.** A comparison of ProQ-like proteins annotated in diverse bacteria (Fig. 1a) identified the 15.5 kDa protein NMB1681 of *N. meningitidis* as a particularly small member, lacking N-terminal or C-terminal sequences that flank the central FinO domain in the well-characterized proteins FinO and ProQ of *E. coli*[15] or RocC of *Legionella*[11] (Fig. 1b). The amino acid sequence of NMB1681 shows sequence conservation within the genus similar to housekeeping proteins in *Neisseria* species (Supplementary Fig. 1 and Supplementary Table 1), arguing that the meningococcal ProQ protein carries out important functions. Similarly, four out of nine residues reported to be essential for RNA regulation by RocC are conserved in the meningococcal ProQ (Fig. 1b).

In the *N. meninigitidis* model strain 8013 used here, the *proQ* gene is expressed as a monocistronic ~473-nt mRNA (according to dRNA-seq data[14]) from the minus strand between *parE* (topoisomerase IV subunit B) and *aroC* (chorismate synthase) (Fig. 1c). Western blot analysis revealed constitutive expression of ProQ in rich media comparable to that of Hfq (Fig. 1d). When grown in rich media (Fig. 1e), a *proQ* deletion strain (strain ΔproQ) shows no difference from wild-type *N. meningitidis*, whereas an *hfq* deletion strain (Δhfq) exhibits a growth defect[16]. However, additional knockout of *proQ* (strain ΔproQ Δhfq) exacerbates the growth defect of the Δhfq mutant (Fig. 1e), which indicates a genetic condition to study ProQ-specific contributions to physiology. Both gene deletions can be fully complemented by 3 × FLAG-tagged *hfq* or *proQ* genes integrated in the *lctP-aspC* locus of strain *N. meninigitidis* 8013, under control of their native promoters (Fig. 1e and Supplementary Fig. 2). Taken together, the observed sequence conservation, expression level and growth phenotype argue for the physiological relevance of ProQ in *N. meningitidis* and likely other *Neisseria* species as well.

**Meningococcal ProQ is a global RBP.** To identify RNA ligands and binding sites of *N. meningitidis* ProQ, we adopted an in vivo UV CLIP-seq protocol[3,4] for analysis of cross-linked RNA-protein complexes after UV light irradiation of live bacteria (Supplementary Figs. 3, 4a). Meningococci expressing C-terminal 3× FLAG tag ProQ from the native locus were grown in rich medium to late logarithmic phase, and cDNA obtained from immunopurified RNA-protein complexes were subjected to high-throughput sequencing (Supplementary Data 1). Comparing the crosslinked samples to a non-crosslinked control, we identified 253 CLIP peaks associated with 166 mRNAs and 16 sRNAs (Fig. 2). The ProQ sites were evenly distributed across both strands of the *N. meningitidis* 8013 genome and included numerous genes important for meningococcal-host interactions (e.g., *iga*, *sodC*, *pilX*, and *dsbA1*) and basic physiology (e.g., *carA*, *carB*, *comE1*, and *pnp*) (Fig. 2a). While most (78%) of the ProQ sites mapped to mRNAs (231 gene associations in 166 distinct mRNAs), we also detected peaks in 16 validated sRNAs[14] (Fig. 2b). Note that peaks can have multiple annotations if they overlap multiple features, e.g., CDSs and 3' UTRs. However, the majority of mRNAs possess only one ProQ CLIP-peak (Fig. 2c), usually in their 3'UTR (Fig. 2a, b), while 66 and 8 binding sites mapped inside a CDS or 5'UTR, respectively (Fig. 2a, b). Of all categories used (sRNA, tRNA, rRNA, 5'UTR, CDS, 3'UTR), sRNAs showed the highest fraction of bound transcripts relative to their number (Supplementary Fig. 4b). Importantly, there was no significant ProQ binding in rRNAs and only 14 CLIP-peaks in tRNAs, despite the much higher expression of these RNAs. Altogether, the CLIP-seq analysis indicates that *N. meningitidis* ProQ possesses a large in vivo targetome comprising nearly 200 mRNAs and sRNAs. This identifies ProQ as the second global meningococcal RBP after Hfq.

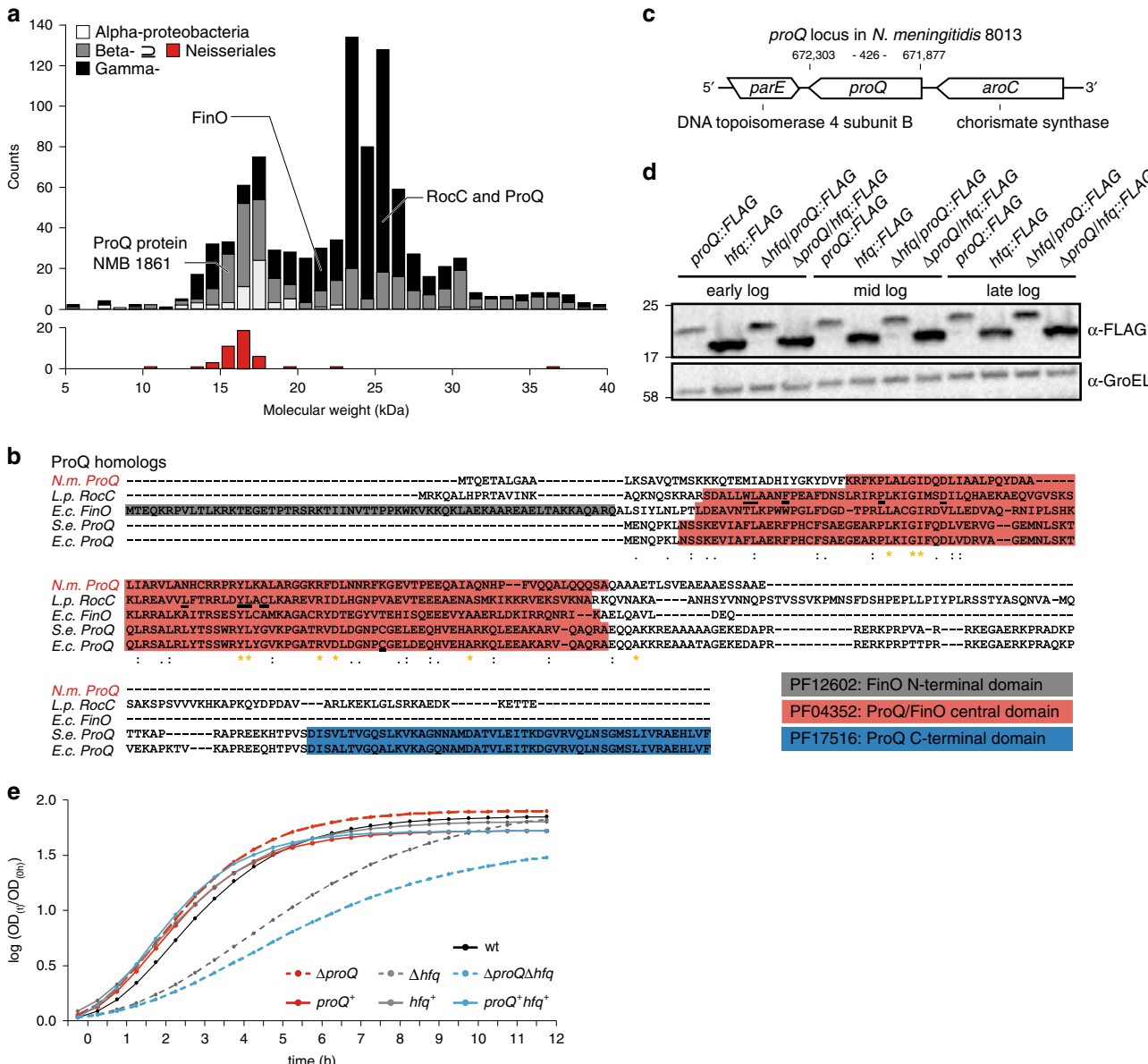

**Fig. 1 Genetic organization and expression of meningococcal ProQ. a** Size distribution of proteobacterial ProQ/FinO-family proteins. Nine hundred thirty nine proteins with the Pfam protein family identifier PF04352 (version 32.0)[67] were included and subdivided into α-proteobacteria, β-proteobacteria, γ-proteobacteria and *Neisseriales*. With a size of 15.5 kDa, the meningococcal ProQ homolog NMB1681 represents one of the smallest ProQ proteins in bacteria lacking additional domains. **b** Amino acid sequence alignment of selected ProQ/FinO-family proteins generated by ClustalW[65]. Abbreviations are as follows: N.m. *Neisseria meningitidis*, L.p. *Legionella pneumophila*, S.e. *Salmonella enterica*, E.c. *Escherichia coli*. The conservation is indicated by asterisks (perfect conservation), colons (conservative substitutions), and periods (semi-conservative substitutions). ProQ/FinO domains (PF04352), FinO N-terminal domains (PF12602) and ProQ C-terminal domains (PF17516) are highlighted in color. Residues required for RNA binding and stabilization in *L. pneumophila*[11] and a thiol-modified cysteine in *E. coli*[38] are underlined. **c** Schematic illustration of the *proQ* locus in *N. meninigitidis* strain 8013. The nucleotide numbers indicate the genomic position in strain 8013. **d** Equal amounts of cells ($OD_{600 nm} = 0.01$) with chromosomally FLAG-tagged *proQ* in the wild-type (wt) and a Δ*hfq* genetic background and chromosomally FLAG-tagged *hfq* in the wt and a Δ*proQ* genetic background were analysed by Western blotting with mouse anti-FLAG antibody (1:1000, Sigma, F1804; anti-mouse-HRP, 1:10,000, Thermo Fischer Scientific, 31430) in three growth phases (early logarithmic, mid logarithmic, and late logarithmic growth phase). GroEL served as control and was detected by anti-GroEL antibody (1:1000, Sigma, G6532; anti-rabbit-HRP, 1:10,000, Thermo Fisher Scientific, 31460). All sizes are given in kilo Daltons (kDa). **e** Growth of *N. meningitidis* wild-type (wt), Δ*proQ*, complemented *proQ*+, Δ*hfq*, complemented *hfq*+, Δ*proQ*Δ*hfq* and complemented *proQ*+*hfq*+ strains in rich medium (GCBL++, see Methods) as determined by measuring the $OD_{600 nm}$ in a 96-well TECAN plate reader. The log-ratio of the $OD_{600nm}$ at time $t$ ($OD_{(t)}$) relative to the $OD_{600nm}$ of the starting culture ($OD_{(0h)}$) is displayed on the y-axis and time $t$ in hours on the x-axis. Source data underlying panels **a**, **d**, and **e** are provided as a Source Data file.

**Meningococcal ProQ binds to structured RNAs.** UV CLIP-peaks often contain crosslink-induced mutations that identify contact regions of RBPs within target transcripts and can help identify RBP binding motifs[17]. Here, we observed cDNA mutations in 28% (67/235) of ProQ peaks, the majority being T→C transitions (Fig. 2d, Supplementary Data 2). We used the MEME suite[18] to search for an RNA motif associated with ProQ binding within a window of 20 nucleotides around

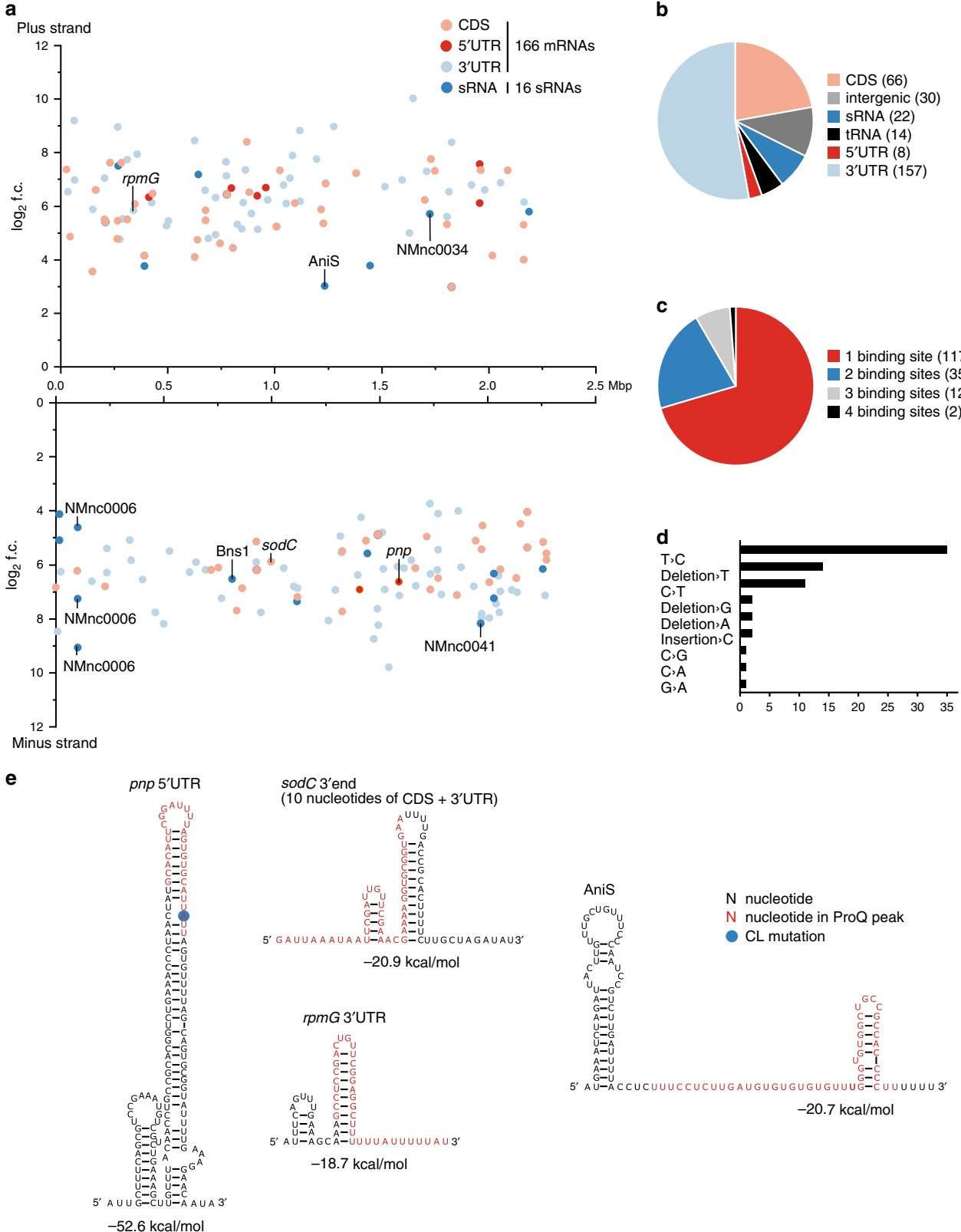

crosslink-induced mutations. However, MEME failed to identify any motif common to the majority ProQ binding regions.

Restricting the MEME search to the 231 ProQ peaks in mRNAs, however, did produce a significant consensus motif contained in 45 sequences (19.5%). This motif occurred in both CDS and 3′UTR peaks and was identical to the neisserial DNA

uptake sequence (DUS) motif ATGCCGTCTGAA[19,20]. Importantly, the genome of *N. meningitidis* strain 8013 contains 1474 exact copies of the DUS and another 682 copies with one mismatch, comprising 1.14% of the genome. There are only 19 DUS sequences in the 166 ProQ-bound mRNAs that do not overlap with a ProQ peak, but given the prevalence of this

**Fig. 2 Properties of ProQ RNA targetome in *N. meningitidis* as revealed by UV CLIP-seq. a** Scatter-plot of CLIP-seq results including 166 mRNAs with their associated annotations (5′ UTR, CDS, 3′UTR), as well as 16 sRNAs that were significantly enriched (log$_2$-fold change (log$_2$ f.c.) ≥ 2 (y-axis), $p_{adj}$-value < 0.05, Benjamini–Hochberg corrected as reported by PEAKachu, https://github.com/tbischler/PEAKachu) in the cross-linked 3× FLAG tagged ProQ strain compared to the non-cross-linked 3× FLAG tagged ProQ. The x-axis represents the genomic position. RNAs selected for further experimental validation are indicated. **b** Pie chart of the ProQ CLIP-seq data showing the relative proportions of ProQ-associated RNA classes. Only significantly enriched RNAs (log2 f.c. ≥ 2; $p_{adj}$-value < 0.05) were included in the analysis. **c** Pie chart showing the distribution of ProQ binding-sites per mRNA based on all segments (5′UTR, CDS, 3′ UTR) of the 166 mRNAs significantly enriched in the CLIP data. **d** Relative abundance of crosslink-specific read mutations. **e** Predicted secondary structures of selected *N. meninigitidis* RNA sequences possessing a ProQ CLIP peak using RNAfold[69]. The ProQ peak sequences are highlighted in color. Source data underlying **b** and **c** are provided as a Source Data file.

sequence in the genome it is difficult to determine whether the observed DUS binding by ProQ is selective or due to chance. We do note that DUS sequences are overrepresented in Rho-independent terminators, so this apparent preference for the DUS could be explained by a preference of ProQ for hairpin structures in 3′UTRs (Fig. 2b, e). Extensive secondary structures were also observed for ProQ sites in mRNA 5′UTRs and in sRNAs, as illustrated in Fig. 2e. This preference of the meningococcal ProQ for RNA structure without a shared primary sequence echoes previous findings with the much larger *Salmonella* and *E. coli* ProQ proteins[4].

**Distinct targetomes of ProQ and Hfq**. We have previously mapped the target suite of Hfq, the only other known global RBP in meningococci, under the same growth condition[14]. Comparing the two targetomes, we observed different RNA binding preferences (Fig. 3a): ProQ binds significantly more frequently in 3′ UTRs than Hfq (62% vs. 13%, $p < 0.001$, two-sample test for equality of proportions with continuity correction), while Hfq shows more binding in CDSes (67% vs. 28%, $p < 0.001$). These differing binding preferences may have mechanistic implications for the effect of these RBPs on their target RNAs.

A COG pathway comparison suggests that Hfq and ProQ each regulate a variety of biological functions (Fig. 3b). Many of the 166 mRNAs with ProQ peaks are involved in energy metabolism (COG C: 10%) and translation (COG J: 10%), out of 194 COG annotated functions. On the other hand, of the 401 Hfq-bound mRNAs[14] over a third code for proteins of unknown function. Together, both RBPs target over a quarter (526) of the meningococcal mRNAs, with limited overlap (41 mRNAs interact with both RBPs, Supplementary Data 3).

A similar investigation of 33 RBP-binding sRNAs supports this view of largely distinct Hfq and ProQ targetomes. Only six sRNAs, including the oxygen limitation-induced AniS[21] and the human blood-induced Bns1[22], show peaks for both RBPs (Fig. 3c). In agreement with previous findings in *Salmonella*[2], ProQ-associated sRNAs tend to have more stable structures (i.e., lower folding free energy when normalized for length) than Hfq-associated sRNAs (Fig. 3d), again indicating that ProQ interactions are structure-driven. In line with observations in *E. coli* and *Salmonella*[4], meningococcal ProQ sRNAs also tend to have shorter poly(U) tails compared to those bound by Hfq (Fig. 3e) although this difference was not statistically significant. This suggests that competition between ProQ and Hfq for the same transcript may involve sequences upstream of the 3′ end.

**ProQ stabilizes mRNAs and sRNAs**. RPBs such as ProQ and Hfq often affect the steady-state levels of their targets by altering RNA stability[1]. To address the effect of the meningococcal ProQ on target stability, we performed northern blot analysis in the same growth condition as used for UV CLIP-seq analysis, selecting the *pnp* mRNA where ProQ binds in the 5′UTR, the *rpmG* RNA showing a ProQ peak in its CDS, and the *sodC*

mRNA where ProQ targets the 3′UTR (Fig. 4a). All three mRNAs showed decreased steady-state levels in the Δ*proQ* strain, as compared to wild-type, and were rescued to wild-type levels by *proQ* complementation (Fig. 4a, Supplementary Figs. 5–7). Similarly, rifampicin stability assays found decreased half-life for each mRNA in the absence of ProQ (Supplementary Figs. 5–7). For instance, the half-life of the *sodC* mRNA declined from 4 min in the wild-type strain to ~2 min in Δ*proQ* (Fig. 4b, Supplementary Fig. 6). Based on these three candidates, the mRNA half-life in the *proQ* mutants was reduced on average by 53% (±17% standard error) compared to the wild-type (Supplementary Figs. 5–7).

Next, we used an electrophoretic mobility shift assay (EMSA) to measure ProQ affinity to these mRNAs in vitro. Each of these three mRNAs shifted in a concentration-dependent manner (Fig. 4d, Supplementary Figs. 5–7). The binding of ProQ to these mRNAs as initially suggested by UV CLIP-seq and confirmed by EMSA, combined with a measurable stability effect, all strongly support that ProQ forms complexes with these transcripts in vivo.

We made similar observations for five selected sRNAs (AniS, Bns1, NMnc0006, NMnc0034 and NMnc0041), finding their steady-state levels were lower in the absence of ProQ (Fig. 5a and Supplementary Fig. 8). Interestingly, NMnc0006 is the only intergenic sRNA that is exclusively associated with ProQ (Fig. 3c). It is an intergenic, highly structured 188-nt sRNA with three detected ProQ sites (Fig. 2a, Supplementary Data 1). Rifampicin stability assays (Fig. 5b, Supplementary Figs. 9, 10) showed a decline in half-life from 4 min to ~1 min in the Δ*proQ* strain, with *proQ* complementation partially restoring it. Overall, we observed decreased stability for all ProQ-bound sRNAs tested in the Δ*proQ* strain (Fig. 5c, Supplementary Figs. 8, 9a, 10a). We also tested several sRNAs by EMSA and consistently observed ProQ association in the high nanomolar range (Fig. 5d, Supplementary Fig. 9c). Importantly, the ProQ-independent sRNA RcoF1 showed no shift with ProQ, confirming the specificity of these interactions (Fig. 5d Supplementary Fig. 9c).

Half of these experimentally confirmed ProQ-bound transcripts are also targeted by Hfq. Whereas *rpmG* levels seem to be raised in the absence of Hfq (Fig. 4a), *sodC* and *pnp* steady-state levels are strongly reduced in both *proQ* and *hfq* deletion strains. Generally, we did not observe a larger decrease in steady-state levels in the absence of both ProQ and Hfq, suggesting redundant functions of Hfq and ProQ on targets that are bound by both RBPs (Fig. 4a). Exceptions include the AniS, Bns1, and NMnc0034 sRNAs whose steady-state levels in the Δ*proQ*Δ*hfq* double-deletion strain were lower than in the respective single deletion strains (Fig. 5a). Unfortunately, their lower steady-state levels in the absence of either RBP alone precludes meaningful half-life comparison with the Δ*hfq*Δ*proQ* strain (Supplementary Figs. 9b, 10b). This notwithstanding, our data show that ProQ directly affects the steady-state levels and the stability of meningococcal mRNAs and sRNAs, largely independently of the other major meningococcal RNA chaperone, Hfq.

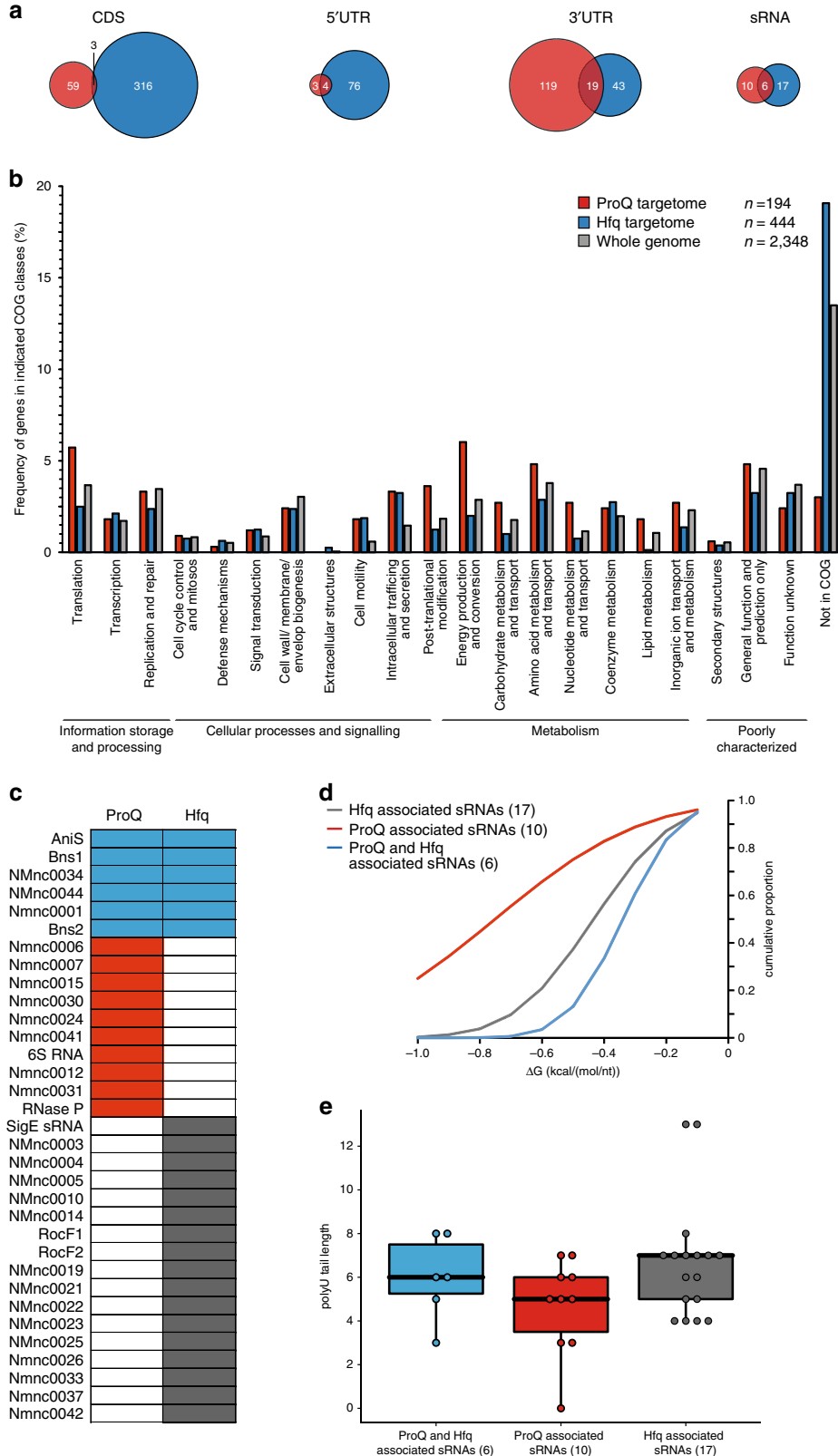

**ProQ globally stabilizes 3′UTRs of mRNAs.** Our in vivo binding map and northern blot validations suggested that ProQ plays a major role in meningococcal gene expression and 3′UTR regulation. To assess this on a genome-wide level and compare RNA-binding with steady-state levels, we subjected the Δ*proQ*

mutant to RNA-seq analysis (Fig. 6, Supplementary Fig. 11, Supplementary Data 4). Absence of ProQ affected gene expression globally, changing the levels of 244 mRNAs, 12 sRNAs and one tRNA (8.56% of genome, Fig. 6a, b, Supplementary Data 4). Although the affected mRNAs code for a wide range of functions,

**Fig. 3 Genome-wide comparison of the ProQ and Hfq targetomes in *N. meninigitidis* strain 8013. a** Venn diagrams comparing the binding locations (CDS, 5′UTR, 3′UTR, sRNA) for ProQ ligands with Hfq ligands. Binding site preferences of Hfq and ProQ differ significantly (Pearson's $\chi^2 = 10.8$, $p < 0.05$). The ProQ CLIP-seq peaks are indicated in red, while Hfq associated transcripts are indicated in blue. **b** Histogram comparing the frequency distribution of mRNAs over the different functional COG classes[62] for mRNAs that bind to ProQ or Hfq[14] as well as for all mRNAs encoded in the *N. meninigitidis* strain 8013 genome. The COG frequency distributions differ significantly for ProQ and Hfq binding mRNAs (Pearson's $\chi^2 = 205.6$, $p < 0.001$). **c** Comparison of all putative *N. meninigitidis* 8013 sRNAs as described in ref. [14]. associated with ProQ according to UV CLIP-seq analysis and/or with Hfq based on RIP-seq analysis described in ref. [14]. **d** Cumulative distributions of predicted length-normalized thermodynamic ensemble folding free energies for ProQ-associated sRNAs, both ProQ-associated and Hfq-associated sRNAs and Hfq-associated sRNAs. **e** Boxplots comparing the polyU tail length distributions between ProQ-associated sRNAs, both ProQ-associated and Hfq-associated sRNAs and Hfq-associated sRNAs (Kruskal Wallis chi squared test, $p = 0.38$). The total number of sRNAs in each group is given in parentheses. The lower and upper hinges in the boxplots correspond to the first and third quartiles. The upper whisker extends from the hinge to the largest value no further than 1.5 * the inter-quartile range (IQR) from the hinge. The lower whisker extends from the hinge to the smallest value at most 1.5 * IQR of the hinge. Data beyond the end of the whiskers are plotted individually. The boxplots were generated with the R packages ggplot2 3.2.1 and ggbeeswarm 0.6.0, and each individual data point is represented by a dot. Source data underlying panels **a**, **b**, **d** and **e** are provided as a Source Data file.

the major impact was seen on amino acid metabolism and transport (COG E) and energy production and conversion (COG C) (Fig. 6c, Supplementary Data 5). Approximately 26% (60/229) of the transcripts with a ProQ peak (Supplementary Data 1 and 4) showed a significant change in steady state levels compared to 8% (272/3383) without a ProQ peak, indicating (i) that ProQ has a significant effect on the expression of bound transcripts, and (ii) numerous secondary effects on the expression of unbound transcripts. These may result from differential expression of global transcription regulators such as 6S RNA[23], $\sigma^E$ sRNA[24] and the *nusA*/*nusB* mRNAs[25], all of which were differentially expressed in the $\Delta proQ$ strain (Fig. 6a, Supplementary Data 4).

Importantly, we also observed that ProQ-dependent changes in transcript levels were associated with particular RNA features from the CLIP-peak analysis (Fig. 6b). While the steady-state levels across all CDSs, 5′UTRs and sRNAs with and without ProQ-binding peaks were not significantly different, 3′UTRs with ProQ-binding peaks had significantly lower steady-state expression levels than 3′UTRs without ProQ CLIP-peaks. In line with this, of the 244 mRNAs with differentially regulated RNA features, 159 mRNAs have downregulated 3′UTR regions (Fig. 6c). To assess whether ProQ can directly protect transcripts from degradation at the RNA 3′end, we cloned and purified recombinant PNPase, a major 3′->5′ exonuclease in *N. meningitidis*, and used it in an in vitro RNA degradation assays. As shown in Supplementary Fig. 12, PNPase efficiently degraded an in vitro transcribed *rpmG* mRNA (whose 3′ end is recognized by ProQ). In contrast, degradation was strongly inhibited when this mRNA was preincubated with ProQ protein. These data indicate that ProQ binding in 3′UTRs may act to stabilize the 3′UTR regions of mRNAs, while ProQ binding in 5′UTRs, CDS and sRNAs generally does not impact RNA steady-state levels.

**ProQ is required for protection against DNA damage.** Previous work on FinO-domain proteins in other species has revealed effects on diverse pathways, and in particular a link to stress response and genomic integrity[8]. Using this as a departure point, we assayed susceptibility of *N. meningitides* $\Delta proQ$ bacteria to oxidative stress as caused by $H_2O_2$. We also included the $\Delta hfq$ mutation, either alone or in combination with $\Delta proQ$. All of these strains showed increased killing by UV light as compared to wild-type (Fig. 7a), and this also extended to killing by $H_2O_2$ (Fig. 7b). However, only the double deletion exhibited significantly impaired survival when stressed with paraquat (Fig. 7c).

In order to comprehensively investigate the influence of ProQ on the life-cycle of meningococci, in particular the transition from meningococcal colonization to invasive infection[26–28], we employed an array of established in vitro an ex vivo virulence assays that comprehensively model all major steps in the disease

process. As depicted in Supplementary Fig. 13, deletion of ProQ had no measurable effect on biofilm formation, adhesion, and invasion to human epithelial cell lines and polysaccharide capsule expression. Therefore, the primary physiological role of ProQ we identified is in protecting against DNA damage, since $H_2O_2$ (via the Fenton reaction[29]) and short wave-length UV light are DNA-damaging agents. As inducing oxidative stress through $H_2O_2$ generation is a major antimicrobial strategy employed by professional phagocytes, ProQ may promote intracellular survival of meningococci and innate immune evasion during invasive disease.

One puzzling observation from these phenotypic assays is that complementation *in trans* was inconsistent, and sometimes failed entirely, which is particularly evident in the $H_2O_2$ condition (Fig. 7b). This heterogeneity in the ability of *trans*-expressed ProQ or Hfq proteins to restore wild-type physiology can also be seen in the steady-state level and RNA half-life experiments above. As we currently lack a good explanation for this, other than that Hfq and ProQ levels may need to be tightly controlled, we believe that this variability is worthy of further investigation.

**Discussion**

The β-proteobacterium *N. meningitidis* is a human-adapted commensal species and a leading cause of epidemic meningitis and sepsis worldwide[26,28,30]. Of known global RBPs in bacteria, it lacks the ubiquitous Csr/Rsm⁻ like RBP but has a functional Hfq with roles in many metabolic and stress pathways[14,22,31] and bacterial survival in human whole blood[13]. The data presented here clearly establish that β-proteobacteria possess another global RBP that not only has a large target suite and defined physiological roles, but also governs its RNA regulon with only a FinO domain. Interestingly, whereas the target spectrum of bacterial Hfq proteins usually comprises dozens to hundreds of different RNA species[3,14,32,33], the targetome of FinO-domain containing proteins seems to be more variable[4,11]. Phylogenetic profiling further showed that all bacterial families without Hfq also lack FinO-domain proteins[8], suggesting that FinO-domain proteins have functions that are complementary to Hfq rather than substituting for this protein. In addition, while a subset of sRNAs and mRNAs are bound by both ProQ and Hfq[2], their binding seems to be mutually exclusive[34].

Echoing the results of recent UV CLIP-seq studies in *E. coli* and *Salmonella*, the β-proteobacterial minimal ProQ protein associates with highly structured sRNAs and binds mRNAs primarily at the 3′UTR. Given that bacterial post-transcriptional gene regulation has generally been associated with the mRNA 5′ UTR (where both translation and RNA decay are initiated), studies of ProQ promise the discovery of new molecular mechanisms associated with the 3′ end of mRNAs. Several

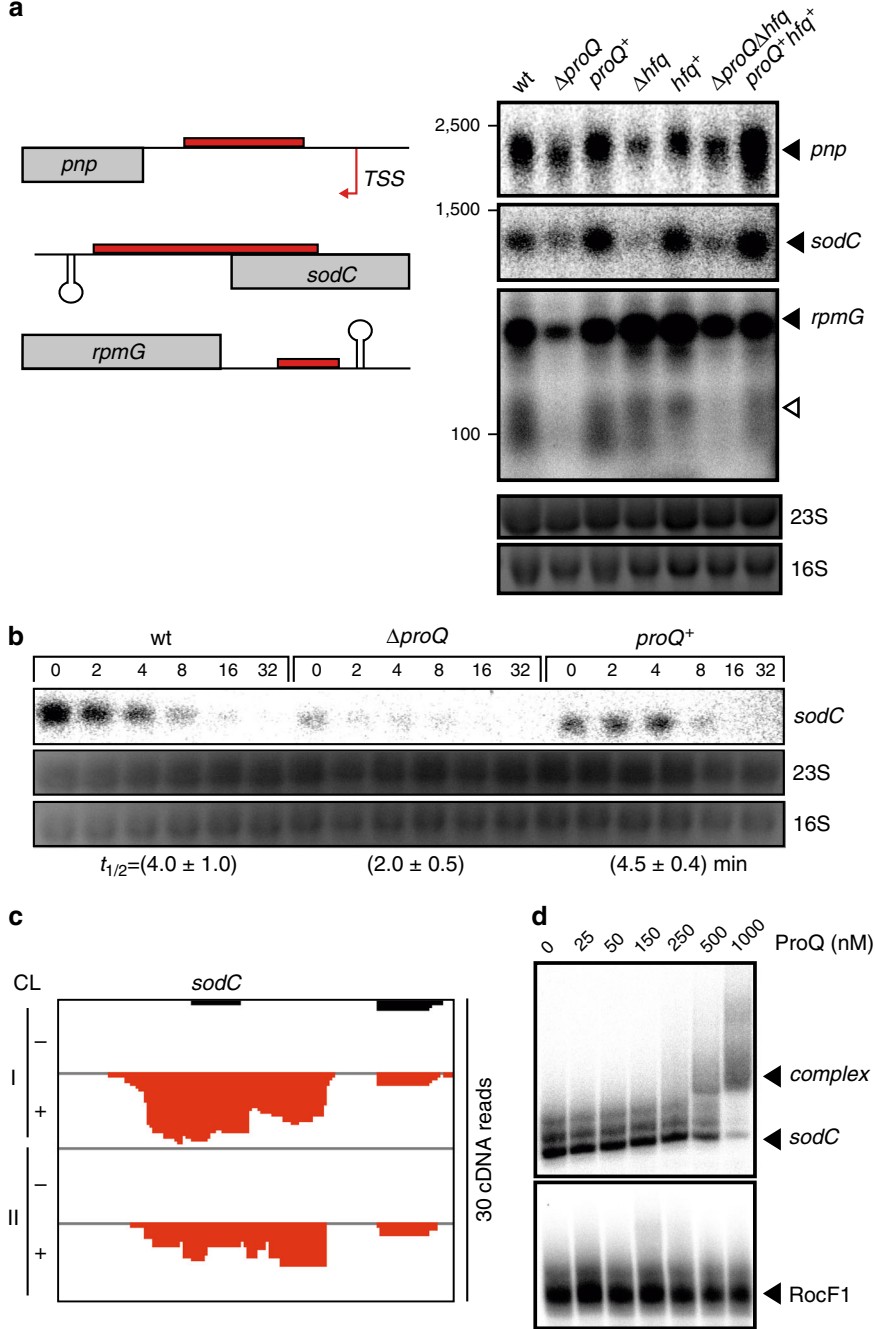

**Fig. 4 Experimental validation of selected ProQ-bound mRNAs revealed by UV CLIP-seq. a** Expression analysis of the ProQ-associated mRNAs *pnp*, *sodC* and *rpmG* derived from UV CLIP-seq peaks (left panel). For the Northern blots (right panel) total RNA was extracted at late logarithmic growth phase (OD$_{600nm}$ of 2.0) from the wild-type strain, the Δ*proQ* mutant strain, the complemented strain *proQ*+, the Δ*hfq* mutant, the complemented strain *hfq*+, the double knock-out strain Δ*proQ*Δ*hfq* and the respective complemented strain *proQ*+*hfq*+ using $^{32}$P-labeled specific DNA probes (see Supplementary Table 4). Filled triangles highlight RNA bands derived from TSS and open triangles highlight bands derived from processing. The housekeeping 16S and 23S rRNAs served as loading controls. **b** Half-life for *sodC* determined in the wild-type, the Δ*proQ* and the complemented *proQ*+ strains, respectively. Northern blots of total RNA extracted at the indicated time points (in minutes) after addition of rifampicin (250 µg ml$^{-1}$) are shown. The housekeeping 16S and 23S rRNAs served as loading controls. The experiments were performed in triplicate and quantifications of RNA half-lives are summarized in Supplementary Figs. 5–7 and 10. **c** Raw read coverage of ProQ UV CLIP-seq peak on *sodC*. Cross-linked samples are indicated by (+), non-cross-linked samples by (−) and the two biological replicates by (I, II). **d** In vitro gel-shift assays of ProQ with *sodC* 3'UTR. Migration of 0.04 pmol in vitro transcribed and $^{32}$P-labeled RNA in a non-denaturing gel after incubation for 20 min with varying concentrations of purified ProQ protein (lane 1–7: 0, 25, 50, 150, 250, 500, 1000 nM). Arrows indicate the RNA-protein complex. An in vitro gel-shift assay with ProQ-independent sRNA RocF1 and increasing concentrations of purified ProQ protein is included as a negative control. Source data underlying panels **a**, **b** and **d** are provided as a Source Data file.

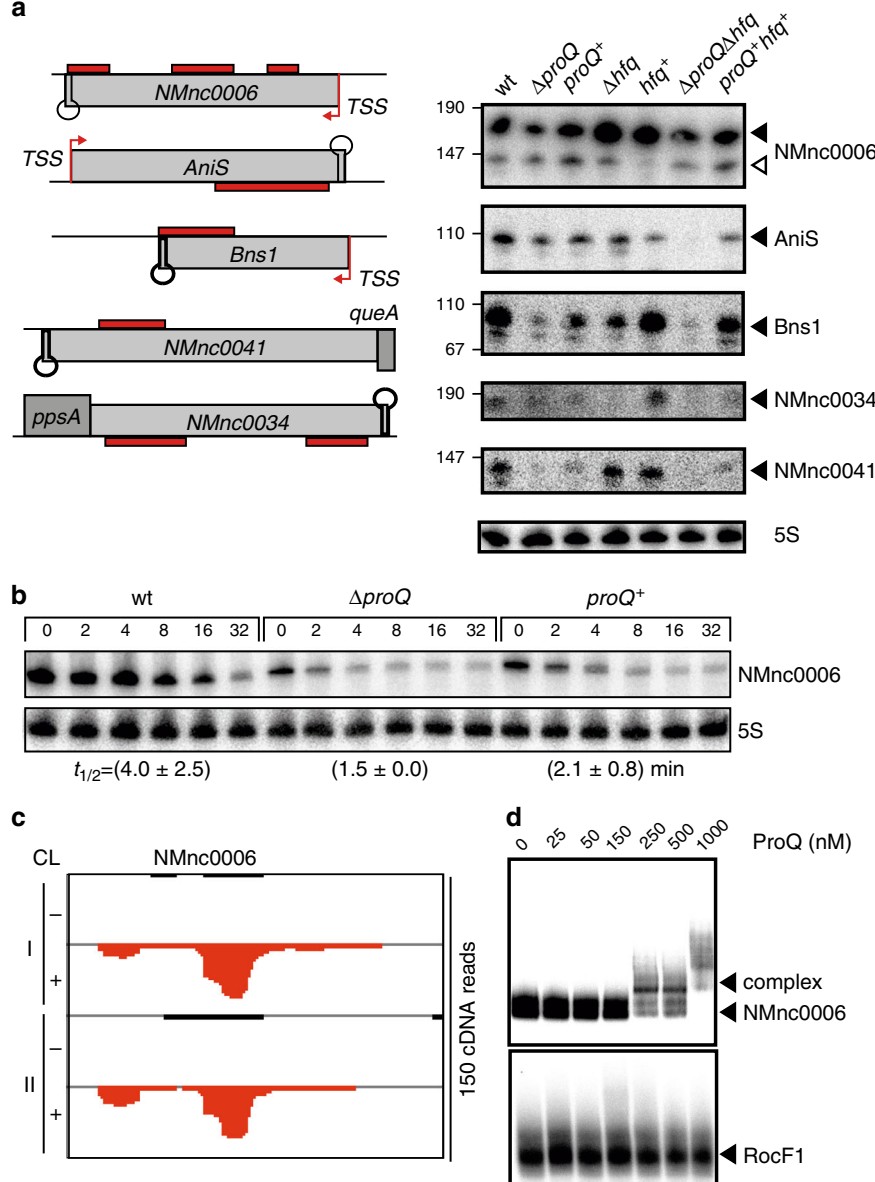

**Fig. 5 Experimental validation of selected ProQ-bound sRNAs revealed by UV-CLIP. a** Expression analysis of the ProQ-associated sRNAs NMnc0006, AniS, Bns1, NMnc0041, NMnc0034 with peak locations (left panel) and corresponding Northern blots (right panel). For the Northern blots (right panel) total RNA was extracted as described in the legend to Fig. 4 and in the Methods section. Filled triangles highlight RNAs bands derived from TSS and open triangles highlight bands derived from processing. Filled triangles highlight RNAs bands derived from TSS and open triangles highlight bands derived from processing. The housekeeping 5S rRNAs served as loading control. **b** Half-life for the sRNA NMnc0006 determined as describd in the legend to Fig. 4b and in the Methods section. The housekeeping 5S rRNAs served as loading control. Experiments were performed in triplicate, and quantifications for RNA half-lives are summarized in Supplementary Figs. 5–7 and 10. **c** Raw UV CLIP read coverage over NMnc0006. Cross-linked samples are indicated by (+), non-cross-linked samples by (−) and the two biological replicates by (I, II). **d** In vitro gel-shift assays of ProQ with NMnc0006 as described in the legend to Fig. 4d and in the "Methods" section. Arrows indicate the RNA-protein complex. In vitro gel-shift assay with ProQ-independent sRNA RocF1 and increasing concentrations of purified ProQ protein as a negative control. Source data underlying panels **a**, **b** and **d** are provided as a Source Data file.

mRNAs with ProQ crosslinks in their 3′UTR such as *rpmG* revealed decreased steady-state level in the *proQ* deletion strain, especially in their 3′UTR regions (Figs. 2b, 4a, 6b, Supplementary Fig. 7), suggesting a role for ProQ in 3′ end-dependent protection from RNA degradation as described for ProQ in *Salmonella*[4]. Indeed, our in vitro results with purified *N. meningitidis* PNPase (Supplementary Fig. 12) support the notion that ProQ can protect against RNA decay at the 3′ end.

In *Salmonella*, ProQ stabilizes some mRNAs by counteracting decay by the 3′ → 5′ exoribonuclease RNase II and endonuclease E by an unknown mechanism, possibly through steric hinderance

of RNase II[4]. In *N. meningitidis*, an RNase E homolog is encoded by NMV_0215 and has been shown to be essential by in vitro transposon mutagenesis[16]. However, nothing is known so far about what role RNase E may play in RNA turnover in *Neisseria*, and we did not find NMV_0215 to be regulated by ProQ (Supplementary Data 1 and 4). While a 3′ → 5′ exoribonuclease RNase II is not annotated in *N. meningitidis*, the mRNA of the widely conserved 3′ → 5′ exoribonuclease *pnp* is targeted by ProQ in its 5′UTR (Fig. 2a, Supplementary Data 1), resulting in higher transcript stability (Fig. 4a, Supplementary Fig. 5). Although *pnp* expression levels could only be partially restored by *proQ*

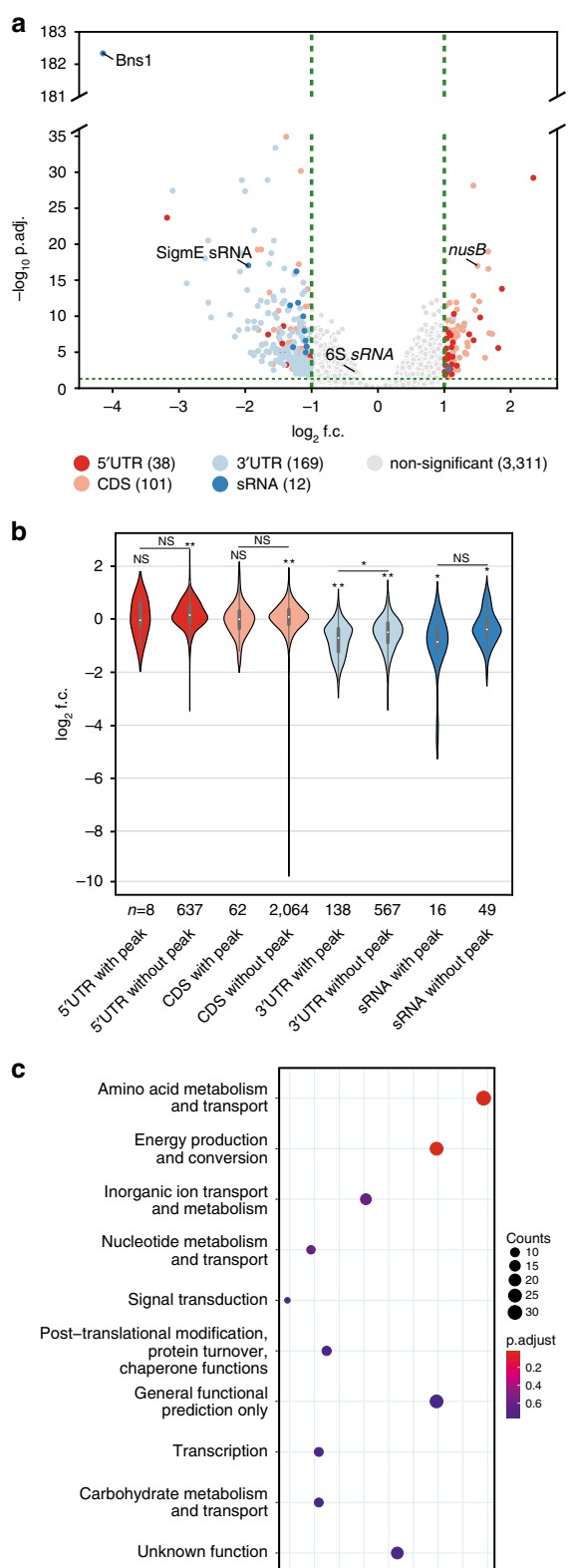

**Fig. 6 Gene expression changes in *N. meninigitidis* mediated by ProQ.**
**a** Volcano plot depicting mean expression changes ($\log_2$ f.c., *x*-axis) of 3631 genomic features excluding *proQ* (NMV_0689) in a $\Delta proQ$ strain compared to the wild-type (wt) plotted against the corresponding $-\log(10)$ adjusted $p_{adj}$-values (*y*-axis), determined by DESeq2[57] at late log phases ($OD_{600\,nm} = 2.0$) in three independent experiments. Features with a log2 f.c. $\geq 1$ and a $p_{adj}$-value $< 0.05$ are highlighted in color. **b** Violin plot showing differential expression changes ($\log_2$ f.c., *y*-axis) across genomic features (*x*-axis), with or without a significant ProQ UV CLIP peak. The black bar in the center of each violin is the interquartile range (IQR). The black lines stretched from the bar depict the lower/upper adjacent values, defined as first quartile minus 1.5 IQR and third quartile plus 1.5 IQR, respectively. *p* values were determined with the Kruskal–Wallis rank sum test ($\chi^2 = 764$, $p = 0.00$) followed respectively by *post hoc* two-sided Dunn's test for the comparison of RNA classes with and without UV CLIP-peak (Bonferroni multiple testing adjustment for $n = 4$) and the Wilcoxon signed rank test for the comparison of each RNA expression level against log2 f.c. = 0 (Bonferroni adjustment for $n = 8$). *$*p < 0.05$; **$p < 0.01$, NS not significant. **c** Functional enrichment analysis based on the COG classification scheme[62] of the $\Delta proQ$ mutant strain compared to the wild-type control based on their fold changes and adjusted *p*-values. The *x*-axis gives the enrichment ratios of the top ten enriched functional classes in the $\Delta proQ$ mutant compared to the wild-type. Significantly enriched COG classes ($p_{adj}$-value. $< 0.05$) are indicated in color. The figure was generated using the "dotplot" function on the "enricher" results object in clusterProfiler[63] v3.10.1. Source data underlying panels **a** and **b** are provided as a Source Data file.

identify a shared Hfq and ProQ targetome of 41 mRNAs and 6 sRNAs (Fig. 3, Supplementary Data 3). Along with the observation that a deletion of ProQ leads to an additive growth phenotype with the *hfq* knockout strain (Fig. 1e) these data suggest regulatory cross-talk between Hfq and ProQ at the post-transcriptional level, and raise the possibility of functional dependence of the affected RNAs on both RBPs (Figs. 4, 5). The current literature is mixed on cross-talk between RBPs in bacteria. For example, in *E.coli* the sRNA McaS regulates biofilm formation through both Hfq and the carbon storage regulation chaperone CsrA[35,36]. However, in *Salmonella* the RaiZ sRNA was initially identified as an Hfq-associated sRNA via a RIP-seq screen[37], but was later shown to depends exclusively on ProQ for both intracellular stability and mRNA target regulation[7]. Further experiments will be necessary to understand the interplay of Hfq and ProQ on neisserial transcripts.

Similarly to previous observations in *E. coli* and *Salmonella*[38–40], we detected reduced DNA damage repair capacity and oxidative stress tolerance in *N. meningitidis* $\Delta proQ$ (Fig. 7). Therefore, regulating genome integrity may indeed be a core function of FinO-domain proteins. Yet, our observed oxidative stress phenotypes were only partially restored in the complemented strains (Fig. 7). This suggests that either the presence of the FLAG-tag interferes with proper functioning of the ProQ protein in the complemented strains and/or that ProQ levels must be tightly controlled for successful DNA damage and oxidative stress repair. Incomplete complementation of ProQ function was indeed reported before for ProQ-dependent host invasion control[5], *rho* mRNA regulation, and sRNA expression[2] in *Salmonella*. Therefore, the success of complementation might be dependent on the abundance of the direct RNA targets as well as associated nucleases.

On the molecular level, we identified direct ProQ target genes that could contribute to the observed phenotypes, such as the superoxide dismutase *sodC* (Fig. 4, Supplementary Fig. 6, Supplementary Data 1). SodC has been reported to protect *N. meningitidis* against oxidative stress[41]. In addition, the direct

complementation (Fig. 4), this RNase is an interesting candidate for a potential regulatory feedback mechanism that measures protein occupancy at mRNA ends (Supplementary Fig. 12).

The association of meningococcal ProQ with 3′UTRs stands in clear contrast with Hfq, which preferentially binds mRNA leader sequences and a distinct set of sRNAs (Fig. 3). However, we did

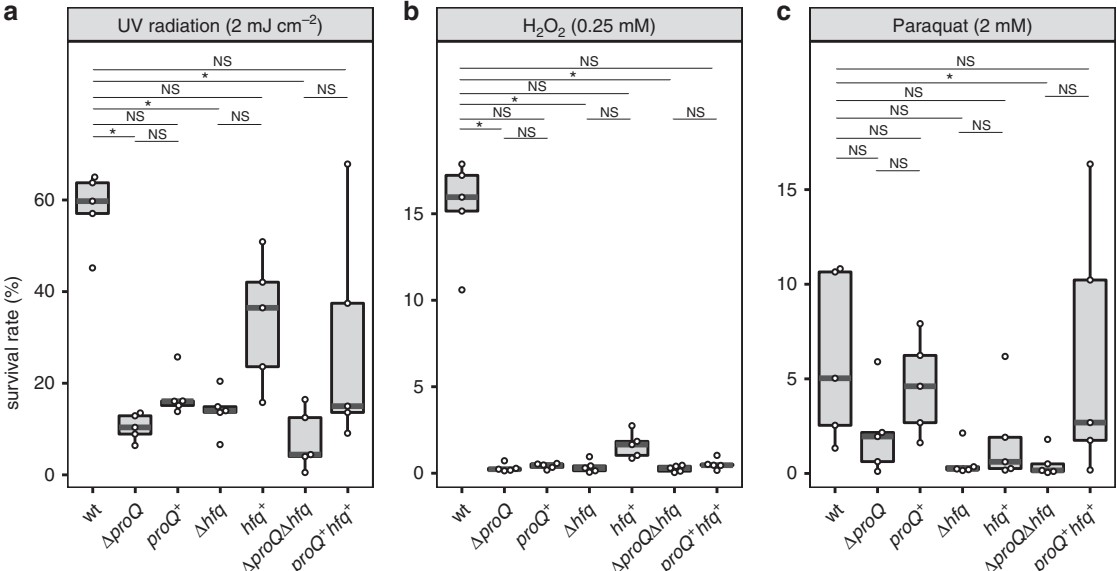

**Fig. 7 ProQ is required for oxidative stress tolerance.** Sensitivity of *N. meninigitidis* wild-type, Δ*proQ*, complemented *proQ*+, Δ*hfq*, complemented *hfq*+, Δ*proQ*Δ*hfq* and complemented *proQ*+*hfq*+ strains to oxidative stress and UV light. In each panel, the data are presented as box-and-whisker plots of five independent experiments, generated with the R packages ggplot2 3.2.1 and ggbeeswarm 0.6.0. The result of each individual experiment is represented by a point. The strains tested are indicated on the *x*-axis and survival relative to untreated controls on the *y*-axis. The lower and upper hinges in the boxplots correspond to the first and third quartiles. The upper whisker extends from the hinge to the largest value no further than 1.5 * the inter-quartile range (IQR) from the hinge. The lower whisker extends from the hinge to the smallest value at most 1.5 * IQR of the hinge. Data beyond the end of the whiskers are plotted individually. *P* values were determined with the Kruskal–Wallis rank sum test followed by two-sided Dunn's test for multiple comparisons with Bonferroni adjustment ($n = 9$) of each mutant against the wild-type and the respective complemented strain. **a** Sensitivity to UV light. Serial dilutions of *N. meningitidis* strains were plated on 5% blood agar plates and exposed to either zero or 20 mJ cm$^{-2}$ of UV of 254 nm wavelength and incubated in 5% $CO_2$ at 37 °C for 20 h. **b** Sensitivity to oxidative stress generated by $H_2O_2$ and **c** paraquat. Bacterial dilutions in GCBL++ were incubated at 37 °C with the indicated concentration of $H_2O_2$ for 15 min and paraquat for 60 min. The number of surviving bacteria was determined by plating serial dilutions. *$P < 0.05$; NS not significant. Source data underlying panels **a**–**c** are provided as a Source Data file.

ProQ target gene *pnp* (Fig. 4 and Supplementary Fig. 5) is known to contribute to processing of UV-light induced double-strand breaks in *Bacillus subtilis* and *E. coli*[42,43], and may function in *N. meningitidis* in a similar manner. Recently, several proteins have been identified that are not damaged by oxidative stress conditions, but rather use ROS-mediated thiol modifications to regulate their function[44]. Reversible oxidation of specific cysteine residues allows these redox-regulated proteins to quickly regulate diverse processes such as protein quality control (Hsp33), gene expression (OxyR) and metabolic fluxes (GapDH)[45–47]. Interestingly, the chromosome-encoded ProQ in *E. coli* has been identified as a redox-sensitive protein upon both $H_2O_2$ and NaOCl stress[38,40]. Of note, the majority of proteins with NaOCl-sensitive cysteines differed significantly from the proteins with $H_2O_2$-sensitive cysteines, indicating that the two physiological oxidants regulate distinct sets of in vivo target proteins[38]. It is tempting to speculate that the constitutively expressed meningococcal ProQ protein is regulated by ROS-mediated thiol modifications induced by several physiological oxidants, thus allowing to quickly regulate global gene expression. However, the thiol-modified cysteine (C88) detected in *E. coli*[38] is not conserved among meningococci (Fig. 1b), though the *N. meninigitidis* ProQ harbors other cysteine residues which may function similarly.

Perhaps the most important results obtained with this minimal FinO-domain protein is that ProQ/FinO-domain proteins are intrinsically global RNA binders that recognize some abundant type of RNA shape. It now appears that binding selectivity evolves through the acquisition of additional N-terminal and C-terminal linkers and domains that narrow target recognition. It is also now clear that there is no correlation between the sizes of ProQ

proteins and the size of their target suites. What then might be driving changes in the domain composition of the ProQ protein? Many γ-proteobacteria including the *Enterobacterales* possess, on average, the largest ProQ proteins (Fig. 1a). Many *Enterobacterales* species inhabit a number of different ecological niches such as soil, water, and the intestines of living organisms that imply exposure to a wide range of environmental conditions with a wide range of differing temperatures. Could this environmental complexity be a driving force in the evolution of more complex ProQ proteins, containing additional domains that modulate the FinO domain's function? In contrast, small ProQ proteins are found in many medically important α-proteobacteria as well as in the β-proteobacteria, including *N. meninigitidis*. These bacteria have rather restricted ecological niches with less varying temperatures. This is well illustrated by *N. meninigitidis*, which has a minimal ProQ protein and a lifestyle restricted to the human nasopharynx.

## Methods

**Bacterial growth and construction of mutant strains**. A list containing all bacterial strains used in this study is provided in Supplementary Table 2. *N. meningitidis* cells were routinely grown at 37 °C in 5% $CO_2$ with 95% humidity on either Columbia blood agar plates (Becton Dickinson) or GC agar plates (22.2 mM glucose, 0.68 mM glutamine, 0.45 mM co-carboxylase, 1.23 mM Fe(NO$_3$)$_3$; all from Sigma) containing 20 µg ml$^{-1}$ erythromicin, 50 µg ml$^{-1}$ kanamycin, or 20 µg ml$^{-1}$ chloramphenicol (final concentration) as required. Liquid cultures were grown in GCBL++ medium in either 50 ml tubes or 96-well microtiter plates at 37 °C with vigorous shaking. Growth in 96-well microtiter plates was monitored using the Infinite F 200 Pro instrument (Tecan) at 37 °C with 3 mm amplitude shaking. Optical density was monitored every 30 min. More details about bacterial growth are given in the Supplementary Methods.

Details about the generation of *Neisseria* mutant strains are listed in Supplementary Methods. Supplementary Table 2 provides a list with all generated

mutant strains which were generated by natural transformation of plasmids or PCR-amplified constructs carrying an antibiotic cassette. A list containing all plasmids utilized for mutant construction is given in Supplementary Table 3 and a complete list of DNA oligonucleotides utilized for mutant construction including the strategy for the combinatory set-up to generate overlap PCRs is provided in Supplementary Table 4. *N. meningitidis* Δ*proQ*, Δ*hfq*, and Δ*proQ*Δ*hfq* and *proQ*::3xFLAG mutant strains were constructed by sub-cloning of plasmids in *Escherichia coli* Top10 cells as described previously[14]. Briefly, the *proQ* gene (NMV_0698) was deleted from 8013 strain by replacing the complete coding sequence by the insertion of the chloramphenicol resistance cassette catGC[48]. The *hfq* gene (NMV_1689) was deleted from strain 8013 by replacing the coding sequence by the insertion of the kanamycin resistance cassette *aphA-1* (GE healthcare). To construct a *proQ*::3xFLAG-tagged strain, a plasmid (3xFLAG:: *aphA-1*) containing the 3xFLAG and the kanamycin resistance cassette *aphA-1* (GE healthcare) flanked by 500 nt upstream and downstream of the *proQ* stop codon was cloned into *E. coli* Top10 as described in the ref.[14]. Transformants were verified by colony PCR on gDNA and in-frame fusion of *proQ*::3xFLAG by sequencing, respectively. All *N. meningitidis* complementation strains were constructed by overlap PCR as described in the refs.[49,14]. The strains were generated by transformation of the obtained overlap PCR fragments into the *lctP* and *aspC* locus of strain *N. meninigitidis* 8013. All PCR products carried fragments encoding the target gene::3xFLAG including its native promotor, the erythromycin resistance cassette *ermC*[50] and ~500 bp of homologous sequences of the complementation locus. Transformants were verified by colony PCR on gDNA.

**Sensitivity to UV irradiation**[51]. Briefly, serial dilutions of *N. meningitidis* strains grown overnight on solid media were plated on 5% Columbia blood agar plates and exposed to 2 mJ cm$^{-2}$ of UV irradiation at 254 nm (UV stratalinker 1800, Stratagene) for 2.5 s while the negative controls (0 mJ cm$^{-2}$) were left in the dark for the same time prior to incubating all plates in 5% CO$_2$ at 37 °C for 20 h[51]. The number of colonies formed by the bacteria was counted with the colony counter ProtoCOL (Meintrup DWS). The survival rates were calculated as the ratio of UV-irradiated survivors to the total number of cells. The assay was performed five times.

**Oxidative stress assays**[52]. Serial dilutions of *N. meningitidis* strains grown overnight on solid media were exposed to H$_2$O$_2$ (0.25 mM) for 15 min and paraquat (2 mM) for 60 min at 37 °C with vigorous shaking prior to plating serial dilutions on 5% Columbia blood agar plates. After incubation in 5% CO$_2$ at 37 °C for 20 h, the survival rates were calculated as the ratio of H$_2$O$_2$ and paraquat-stressed survivors to the total number of cells[52]. The number of colonies formed by the bacteria was counted with the colony counter ProtoCOL (Meintrup DWS). The assays were performed five times.

**UV crosslinking and immunoprecipitation (UV-CLIP)**. In two replicate experiments, *N. meningitidis proQ*::3xFLAG mutant strains expressing 3 × FLAG-tagged ProQ protein were grown in liquid GC medium containing kanamycin until an OD$_{600nm}$ of 2.0 representing late logarithmic growth phase. For each strain, cells equivalent to an OD$_{600nm}$ of 200 were collected and subjected to UV CLIP as previously described[3]. Half of the culture was directly placed in a 22 by 22 cm plastic tray and irradiated with UV light at 800 mJ cm$^{-2}$ at 254 nm while the other half of bacterial cultures was stored in 50 ml Greiner tubes on ice. Cells were pelleted in 50 ml fractions by centrifugation for 40 min at 6000 × g and 4 °C and resuspended in 800 μl NP-T buffer (50 mM NaH$_2$PO$_4$, 300 mM NaCl, 0.05% Tween, pH 8.0). The resuspended pellets were mixed with 1 ml glass beads (0.1 mm radius), 20 μl lysozyme (25 mg ml$^{-1}$, Roth) and 10 μl DNaseI (Fermentas) and incubated at 37 °C for 10 min at 900 rpm shaking. After cooling the samples for 2 min on ice, the cells were lysed using a Retsch MM40 ball mill (30 s$^{-1}$, 10 min) in pre-cooled blocks (4 °C) and centrifuged for 15 min at 16,000 × g and 4 °C. Cell lysates were transferred to new tubes and centrifuged for 15 min at 16,000 × g and 4 °C. The cleared lysates were mixed with one volume of NP-T buffer with 8 M urea followed by incubation for 5 min at 65 °C in a thermomixer with shaking at 900 rpm and dilution in 10× in ice-cold NP-T buffer. Meanwhile, anti-FLAG magnetic beads (Sigma) were washed three times in NP-T buffer (30 μl 50% bead suspension was used for a lysate from 100 ml bacterial culture). The washed glass-beads were mixed with the cleared lysate and incubated for 1 h at 4 °C while rotating before the beads were collected by centrifugation at 800 × g. After resuspending the beads in 1 ml NP-T buffer, they were transferred to new tubes, washed once with high-salt buffer (50 mM NaH$_2$PO$_4$, 1 M NaCl, 0.05% Tween, pH 8.0) and twice with NP-T buffer. Ten percent of the beads from each sample were resuspended in 15 μl 1× PL (0.3 M Tris-HCl pH 6.8, 0.05% Bromophenol blue, 10% Glycerol, 7% DTT) for western blot analysis of the unlabeled eluates. After one wash with CIP buffer (100 mM NaCl, 50 mM Tris–HCl pH 7.4, 10 mM MgCl$_2$), the beads were resuspended in 100 μl CIP buffer with 10 units of calf intestinal alkaline phosphatase (NEB) and incubated for 30 min at 37 °C in a thermomixer with shaking at 800 rpm. After one wash with high-salt buffer and two washes with PNK buffer (50 mM Tris–HCl pH 7.4, 10 mM MgCl$_2$, 0.1 mM spermidine), the beads were resuspended in 1 ml NP-T buffer and stored at 4 °C overnight. After washing the beads with 1 ml PNK buffer, the remaining beads were resuspended in 100 μl

PNK buffer with 10 U of T4 polynucleotide kinase (Fermentas) and 10 μCi γ-$^{32}$P-ATP and incubated for 15 min at 37 °C. The beads were washed three times with NP-T buffer prior to resuspending the beads in 20 μl Protein Loading buffer (0.3 M Tris–HCl pH 6.8, 0.05% bromophenol blue, 10% glycerol, 7% DTT) and incubation for 5 min at 95 °C with occasional vortexing. The magnetic beads were removed on a magnetic separator and the supernatant was separated on a 15% SDS–Tris-glycine (pH 6.8/8.8) polyacrylamide gel system. RNA–protein complexes were transferred to a nitrocellulose membrane, the protein marker was highlighted with a radioactively labeled marker pen and exposed to a phosphor screen overnight. The autoradiogram was aligned on the membrane to cut out the labeled RNA–protein complexes from the membrane. Each membrane piece was further cut into smaller pieces, which were incubated in 400 μl PK solution [50 mM Tris–HCl pH 7.4, 75 mM NaCl, 6 mM EDTA, 1% SDS, 10 U SUPERaseIN (Life Technologies), and 1 mg ml$^{-1}$ proteinase K (ThermoScientific)] for 30 min in a thermomixer at 37 °C with shaking at 1100 rpm. The incubation was continued for additional 30 min after adding 100 μl 9 M urea. The tubes with the membranes were spun down and about 450 μl of the PK solution/urea supernatant was mixed with 450 μl phenol:chloroform:isoamyl alcohol in a phase-lock tube and incubated for 5 min in a thermomixer at 30 °C with shaking at 1,000 rpm. After centrifugation for 12 min at 16,000 × g and 4 °C, the aqueous phase was precipitated with three volumes of ice-cold ethanol, 1/10 volume of 3 M NaOAc pH 5.2, and 1 μl of GlycoBlue (Life Technologies) in LoBind tubes (Eppendorf). The precipitate was centrifuged for 30 min at 16,000 × g and 4 °C and the resulting pellets were washed with 80% ethanol, centrifuged again for 15 min at 16,000 × g and 4 °C, dried 2 min at room temperature and resuspended in 6 μl sterile water. The samples were stored at −20 °C.

**RNA extraction for Northern blot analysis and RNA-seq**. Samples were collected from bacterial cultures grown in GCBL$^{++}$ liquid medium to an OD$_{600nm}$ of 2.0, corresponding to the late logarithmic growth phase. After fixing the samples by addition of STOP Mix [95% (vol/vol) EtOH and 5% (vol/vol) phenol], the pelleted cells were frozen in liquid nitrogen and stored at −80 °C until RNA extraction. For RNA extraction frozen bacterial cultures were thawed on ice and centrifuged for 10 min at 4000 × g at 4 °C. Cell pellets were resuspended in a lysis solution consisting of 600 μl of 0.5 mg ml$^{-1}$ lysozyme in TE buffer (pH 8.0) and 60 μl 10% SDS. Bacterial cells were lysed by heating the samples for 1–2 min at 65 °C. The lysates were used for total RNA extraction according to the hot phenol method[53].

**cDNA library preparation and sequencing for UV CLIP-seq and RNA-seq**. cDNA libraries of RNA-seq-samples were constructed by Vertis Biotechnology AG, Munich, Germany. To deplete ribosomal transcripts, RNA-seq samples were treated with the Ribo-Zero "Bacteria" kit (Illumina) followed by RNA fragmentation and adapter ligation. UV CLIP-seq libraries were prepared using the NEBNext Multiplex Small RNA Library Prep Set for Illumina (#E7300, New England Biolabs) according to the manufacturer's instructions as described before[3]. cDNA libraries of RNA-seq samples were pooled on an Illumina NextSeq 500 high-output flow cell and sequenced in single-read mode (1 × 76 cycles). cDNA libraries of UV CLIP samples were pooled on an Illumina NextSeq 500 mid-output flow cell and sequenced in paired-end mode (2 × 75 cycles). The raw, de-multiplexed reads as well as the normalized coverage files of all cDNA libraries have been deposited in the National Center for Biotechnology Information's Gene expression omnibus (GEO) and are accessible via the GEO accession number GSE129868. Statistics on cDNA library sequencings are summarized in Supplementary Tables 5 and 6.

**Processing of UV CLIP-sequence reads and mapping**. Bioinformatic analysis of the resulting UV CLIP-reads followed the protocol established previously[4]. In brief, read quality was assessed using FastQC version 0.11.2. Paired end reads were quality trimmed independently of each other for a Phred score cutoff of 20 using fastq_quality_trimmer from the FASTX toolkit verion 0.0.13. After quality trimming, the NEBNext R1 and R2 adapters (R1: AGATCGGAAGAGCACACGTCTG AACTCCAGTCAC, R2: GATCGTCGGACTGTAGAACTCTGAACGTGTAGAT CTCGGTGGTCGCCGTATCATT) were trimmed using cutadapt v1.71[54]. Putative PCR duplicates were removed using FastUniq[55]. The remaining reads were mapped to the *Neisseria meningitidis* 8013 genome (NCBI accession: NC_017501.1) using READemption version 0.3.7 and segemehl version 0.2.0 with an accuracy cutoff of 80%, and only uniquely mapping reads were retained for further analysis. NCBI annotations were supplemented with UTR and sRNA annotations previously defined using transcriptional start site mapping and computational prediction of Rho-independent terminators[14,56].

**Peak calling (UV CLIP)**. Normalization and peak calling was performed using a similar procedure to that previously described[4]. Briefly, an exploratory analysis showed a clear bimodal distribution of the log ratio of read counts between the cross-linked and non-cross-linked library read counts. Read counts over positions in the mode near 0 (i.e., approximately equal read counts) were used to calculate normalization factors using the geometric mean normalization procedure introduced by DESeq[57]. Peak calling was then performed using the adaptive approach of PEAKachu (https://github.com/tbischler/PEAKachu, manuscript in preparation)

with paired-end (-P) and paired-replicates (-r) options set, using the manually set normalization factors described above. Peaks with a $\log_2$-fold change ($\log_2$ f.c.) greater than 2 and adjusted P-value ($P_{adj}$-value, Benjamini–Hochberg corrected) less than 0.05 were considered significant in our analysis.

**Analysis of crosslink-specific mutations (UV CLIP).** Detection of crosslink-induced cDNA mutations was done following our previous approach[3]. Mutated sites were required to be present in both paired reads mapping to a significant peak, and the first read was extracted using samtools before applying PIPE-CLIP with a false discovery rate cut-off of 0.1 and requiring at least ten reads mapped. Any putative crosslinking induced mutations also present in non-crosslinked samples were eliminated.

**Processing of sequence reads and mapping (RNA-seq).** Illumina reads were quality and adapter trimmed with Cutadapt [version 1.16 using a cutoff Phred score of 20 in NextSeq mode and reads without any remaining bases were discarded (command line parameters: --nextseq-trim=20 -m 1 -a AGATCGGAAGAGCACACGTCTGAACTCCAGTCAC)]. After trimming, we applied the pipeline READemption[58] version 0.4.5 to align all reads longer than 11 nt to the *N. meningitidis* 8013 (NCBI accession: NC_017501.1) genome using segemehl version 0.2.0[59] with an accuracy cut-off of 95%.

**Differential expression analysis (RNA-seq).** We applied READemption to quantify read alignments for each library with the same annotations used in the UV CLIP-seq analysis (treating pseudogenes as CDSs and excluding miscRNAs) on the sense strand. Reads overlapping genomic features by at least 10 nt were counted as mapped to that feature. If the read overlapped more than one annotation, the value was divided by the number of annotationss and counted separately for each annotation (e.g., 1/3 for a read mapped to 3 annotations). Differential expression analysis comparing Δ*proQ* to wild-type samples was done with DESeq2[60] version 1.22.1. Fold-change shrinkage was applied by setting the parameter betaPrior to TRUE. All features with $\log_2$ f.c. $\leq -1$ or $\geq 1$ and adjusted P-value < 0.05 were considered significantly differentially expressed.

**Functional enrichment analysis (RNA-seq).** To identify functional classes overrepresented in differentially expressed protein-coding genes in the comparison of Δ*proQ* and wild-type strains, we applied functional enrichment analysis based on COG classes[61,62] using the function "enricher" of the R package clusterProfiler[63] v3.10.1. The analysis was conducted separately for all regulated genes, only upregulated genes ($\log_2$ f.c. $\geq 1$) and only downregulated genes ($\log_2$ f.c. $\leq -1$). Locus tags of positive genes in each case were identified as the union of regulated CDSs, 5′UTRs and 3′UTRs, meaning that a only single feature associated with a coding sequence was required to be regulated to be included in the analysis. Locus tags of all features not present in COG were assigned to an artificial class X (not in COG).

**Rifampicin RNA stability assay.** Bacterial cells grown to $OD_{600nm}$ 2.0 were treated with rifampicin (final concentration 500 µg ml$^{-1}$). RNA samples were taken at indicated time points (0, 2, 4, 8, 16, and 32 min) and were immediately mixed with 1/5 volume of 95% ethanol and 5% phenol and frozen in liquid nitrogen. RNA was isolated using the hot phenol method[3] and RNA stability was analyzed by Northern blotting using ImageQuant Tools from AIDA software (Raytest, Germany) for quantification of the RNA expression intensities.

**Northern blot analysis.** For Northern blot analysis of sRNAs, 5 µg of DNase I-treated total RNA was separated on a 6% polyacrylamide (PAA) gel containing 8.3 M urea. PAA gels were either stained with ethidium bromide or RNA was transferred onto Hybond-XL membranes (GE Healthcare). To identify mRNAs, 10 µg of total RNA was separated on 1.2% agarose gel prior to staining the gel with ethidium bromide to ensure RNA quality and to measure loaded RNA amounts. Afterward the separated RNA was transferred onto Hybond-XL membranes (GE Hhealthcare) by capillary transfer overnight. After cross-linking, the membranes were hybridized overnight at 42 °C with $\gamma^{32}$P-ATP end-labeled oligodeoxyribonucleotide (Supplementary Table 4). Signals were visualized on a Phosphorimager (Typhoon FLA 7000, GE Healthcare) and quantified with the AIDA software (Raytest).

**Electromobility shift assays.** The *N. meninigitidis* 8013 *proQ* gene was cloned into pTYB11 plasmid (NEB) for intein-based expression and purification of a tagless ProQ protein. A detailed purification protocol is given in the Supplementary Methods. DNA templates containing a T7 promoter sequence were generated by polymerase chain reaction and served as templates for in vitro transcription. Oligos used to generate the individual DNA templates are listed in Supplementary Table 4. T7 transcription was carried out with the MEGAscript® T7 kit (Ambion) according to the manufacturers' protocol. Gel-shift assays were performed as previously described[8]. In short, 5′-end labeled RNA was

denatured (1 min at 95 °C) and cooled (5 min on ice) before adding 1 µg yeast RNA and 10× RNA structure buffer (Ambion). Afterwards, ~0.04 pmol 5′-end labeled RNA (4 nM final concentration) were mixed with increasing amounts of purified ProQ protein (30 nM to 1 mM final concentration) in 10 µl reactions in 1× RNA structure buffer (Ambion). After incubation for 20 min at 37 °C, samples were directly loaded after addition of 3 µl 5× native loading dye (0.5× TBE, 50% (vol/vol) glycerol, 0.2% (wt/vol) xylenecyanol and 0.2% (wt/vol) bromophenol blue) to a native 8% PAGE in 0.5× TBE % (vol/vol). Gel electrophoresis was performed in 0.5× TBE buffer at 300 V. Afterwards, gels were dried for 45 min and analysed using a PhosphoImager (FLA-3000 Series, Fuji) and ImageQuant Tools from AIDA software (Raytest).

**Western blot analysis and SDS-PAGE.** Bacterial cells grown to different growth phases were pelleted by centrifugation at $16,100 \times g$ at room temperature for 2 min and dissolved in Laemmli protein loading buffer (62.5 mM Tris pH 6.8, 2% SDS, 10% glycerol, 5% 2-mercaptoethanol, 0.001% bromphenolblau). After heating for 5 min at 95 °C, 0.01 $OD_{600}$ equivalents of samples were separated by 12% (vol/vol) SDS-PAGE (Tris-glycine chemistry) and either stained with Coomassie 2250 (Serva, Heidelberg) or transferred to a PVDF membrane by semidry blotting as described in the ref. [64].

**Sequence retrieval and alignments.** The UniProt database[65] was used to retrieve information for sequence alignments of the following ProQ sequences (accession numbers are given in parentheses): chromosome-encoded NMB1681 of *N. meningitidis* (Q9JY98); F plsmid encoded FinO of *E. coli* (P29367); chromosome-encoded RocC of *L. pneumophila* (A0A128QHZ1); chromosome-encoded ProQ of *E.coli* (P45577); chromosome-encoded ProQ of *S. enterica* (A0A0H3NC79). Alignments were created using Clustal[66]. For the size comparison of ProQ/FinO-domain proteins among α-proteobacteria, β-proteobacteria, γ-proteobacteria and *Neisseriales*, 939 proteins with the Pfam protein family identifier PF04352 (version 32.0)[67] were included.

**Target suite comparisons of ProQ and Hfq.** Differences in RNA binding preferences of ProQ and Hfq were calculated with two sample test for equality of proportions with continuity correction. Pearson's Chi-squared test without Yate's continuity correction as implemented in R version 3.1.1 (R Development Core Team 2013, https://www.r-project.org/) was used to compare the distribution of genes over the different COG functional classes[61] between ProQ and Hfq bound mRNAs[68].

**Analysis of sequence and structure motifs.** The sequences of significant ProQ peaks ($\log_2$ f.c. > 2, $p_{adj}$-value < 0.05) situated in 3′UTRs and CDSs were used for sequence motif identification with the MEME program[18] with all parameters set at default values with the exception of Motif search on the given strand only and sequence length limitation of 6–20 nucleotides. RNAfold[69] was used to predict secondary structures of selected *N. meningitidis* RNA sequences possessing a ProQ CLIP peak.

**Reporting summary.** Further information on research design is available in the Nature Research Reporting Summary linked to this article.

## Data availability
UV CLIP data and RNA-seq data supporting the findings of this study have been deposited at Gene Expression Omnibus under accession no. GSE129868. The source data underlying Fig. 1a, c, d, e, 2b, c, 3a, b, d, e, 4a, b, d, 5a, b, d, 6a, b, 7, and Supplementary Figs. 2, 3, 4b, 5b–d, 6b–d, 7b–d, 8, 9a–c, 10a, b, 12, 13a–f are provided as a Source Data file. All data is available from the corresponding author upon reasonable request. Source data are provided with this paper.

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

## Acknowledgements

We thank Maren Bleckmann for purification of the ProQ protein. We thank Helene Mehling and Barbara Conrad for assisting the laboratory work. We thank the Helmholtz Institute for RNA-based Infection Research (HIRI) who supported this work with a seed grant through funds from the Bavarian Ministry of Economic Affairs and Media, Energy and Technology (Grant allocation nos. 0703/68674/5/2017 and 0703/89374/3/2017). In addition, this work was supported by Interdisciplinary Center for Clinical Research Würzburg Grant IZKF Z-6 (to T.B.).

## Author contributions

S.B. contributed to project design, experimental work, data analysis and interpretation and drafting of the manuscript. M.G. carried out size distribution analysis of ProQ/FinO-domain proteins, and produced recombinant PNPase. N.H. contributed to project design and experimental work. T.B. carried out RNA-seq analysis and developed associated figures. L.B. performed computational analysis of UV-CLIP data, and contributed to manuscript review and editing. C.S. is the corresponding author and provided assistance on statistical analyses, COG analyses and conception, data interpretation, drafting and writing the manuscript. J.V. is the corresponding author and oversaw the project conception and design, data interpretation and writing the manuscript.

## Competing interests

The authors declare no competing interests.
