## [Peer Review File · Nature Communications]

Reviewers' comments:

Reviewer #1 (Remarks to the Author):

In the manuscript entitled "The minimal ProQ protein of *Neisseria meningitidis* reveals an intrinsic capacity for structure-based global RNA recognition by the FinO domain", Bauriedl and colleagues present a study on the RNA-binding protein ProQ from *Neisseria meningitidis*. The authors investigate the target RNAs using CLIP and identify nearly 200 RNAs; some of these are further analysed and used for validation experiments. The authors conclude that ProQ is a global RNA regulator in *Neisseria* that binds mainly in the 3'UTR of mRNA to structured regions and is beneficial in response to oxidative stress and DNA damage.

The manuscript is very well written and the line of thought is clearly laid out. The experimental approach is state of the art and, to the best of my understanding, well conducted.

Major criticism

1. The authors find that ProQ targets are protected from degradation when comparing RNA half-lives of wt and delta proQ mutants. They conclude from this finding that ProQ may counteract decay when binding to the 3'UTR of mRNAs. However, the presented data do not allow to assess if the RNA stabilising effect is really a specific or rather a secondary effect. As stated by the authors, also transcripts of cellular RNases are targets of ProQ (Pnpase) which makes it possible that the observed effects are global and not specific to ProQ binding. A relatively straightforward way to test this would be to select the ProQ binding site in 1-3 of the validation RNAs (sodC, pnp, rpmG, AniS,....) that was found in the CLIP-Seq peaks and replace it with a sequence that is not bound by ProQ. A repetition of the expression levels or half-lives after rifampicin treatment in wt or proQ ko (Fig. 4) would then allow to answer this question. This point must be addressed before suggesting ProQ having "a role in 3' end-dependent protection" (l.304).

2. I am somewhat concerned about the experimental setup of the UV radiation experiment (comp. Fig. 6).

a) l.413/l.959: Why is the energy (20J) given in m^2 but the UV irradiation for the CLIP experiment in cm^2 ?

b) 20J is a very high amount of energy, especially if one wants to squeeze that out of a Stratalinker as described. It must take several minutes to achieve such high irradiation dosages. How does this compare to the average turnover of the validation RNAs that were observed earlier? Is the treatment taking significantly longer than the effect on RNA you described earlier? In this respect, the details might matter a lot: was the negative control (0 J) treated in the same fashion (plated on a tray and left in the dark for the same time)? Please describe this in more detail in the Methods section.

c) l. 430: "Half of the culture was directly placed..." What happened to the other half?

Minor comments

3. l.117: Please add here that the chromosomal gene was "replaced/extended" with a version that has a FLAG tag (= no endogenous, non-tagged version of ProQ; not overexpressed from a plasmid). Its in the

Methods section but stating it here more explicitly would make it much easier to read.

4. l.252: You did not provide any experimental evidence that ProQ influences DNA damage repair in any fashion. Please provide such data or rephrase.
5. l.459: Please specify which type of SDS PAGE you used. Lämmli gels have been proven problematic for RNAs due to the high pH and using this system can lead to degradation of RNA.
6. l.506: Typo: segemhel
7. l. 519: adjusted p-value. How was it adjusted?
8. Fig. 2a: I do not see the value in displaying the ProQ peaks over the genomic distribution. Is there any reason in *Neisseria* why it should bind to only distinct regions? Please clarify.
9. In the same figure: why are the peaks shown for p 0.05 but the pie chart is using p 0.01?
10. In the same figure: The pie chart caption reads "RNA classes" but you show CDS/UTRs with sRNA and tRNA. That's a bit confusing since the former are all mRNA features.
11. Fig. 3c: The y Axis is labeled "polyA" while I assume it should be "polyU". Also, I am not convinced that this data which is not statistically significant at all (l.179) is necessary to support your story.
12. Fig. 6: The labeling (a,b,c) of the panels and the text in the caption is all mixed up. Also, it is not clear if what happened to the 0 and 10J/m² data (they are not mentioned in the Methods part; see comment #2).

Reviewer #2 (Remarks to the Author):

The manuscript by Bauriedl et al. concerns important topic of RNA recognition by the proteins with a ProQ/FinO domain, which are phylogenetically conserved in numerous β - and γ -proteobacteria bacteria, including human pathogens. Proteins of this family often have N- or C-terminal accessory domains besides the homologous ProQ/FinO domains. The RNA binding of proteins with a C-terminal extension (ProQ from *E.coli* and *S.enterica*, and RocC from *L. pneumophila*) or N- terminal extension (F-like plasmid FinO) have already been described. The manuscript by Bauriedl et al is the first that describes the RNA interactome of a protein composed solely of the ProQ/FinO domain. Comparing the RNA ligands of the single-domain *N. meningitidis* NMB1681 protein with those of two-domain ProQ or RocC provides the opportunity to elucidate what is the role of the ProQ/FinO domain in proteins from this family. The analysis of the structures of RNAs detected using the CLIP-seq study allowed the authors to conclude that the ProQ/FinO domain alone has a capability to recognize a subset of RNA molecules in the cell. This conclusion is important because it provides a new mechanistic insight into the function of the ProQ/FinO domain proteins, and it also allows to better understand the role of the ProQ homolog in *Neisseria*. Additionally, the authors characterized the physiological role of this protein, suggesting that its functions are related to those of Hfq, and indicating the physiological processes that are most affected by the ProQ homolog in *Neisseria*. Because of the general importance of the conclusions, this manuscript will be interesting to the wide readership of *Nature Communications*, and hence it is appropriate for the publication.

Specific comments

- 1) What is the likelihood of detecting RNAs that are not direct ligands of ProQ using CLIP-seq? For

example, if an RNA was drawn into proximity of ProQ by pairing with another RNA, which is a specific ligand, could it also be crosslinked and hence identified as a ligand using this method? Could this issue affect the outcome of the analysis?

2) In Fig. 3e the plot is described as polyA distribution, instead of polyU distribution

3) The free RNAs in the binding assays on Fig. 4 are quite heterogenous. Could this correspond to alternative conformers?

Minor comments

Could you clarify the sentence on page 6, lines 168, 169?

In legend to figure 4a, did you mean “horizontal” instead of “vertical orange bar”?

Mikołaj Olejniczak

Comments to reviewer #1:

In the manuscript entitled "The minimal ProQ protein of *Neisseria meningitidis* reveals an intrinsic capacity for structure-based global RNA recognition by the FinO domain", Bauriedl and colleagues present a study on the RNA-binding protein ProQ from *Neisseria meningitidis*. The authors investigate the target RNAs using CLIP and identify nearly 200 RNAs; some of these are further analysed and used for validation experiments. The authors conclude that ProQ is a global RNA regulator in *Neisseria* that binds mainly in the 3'UTR of mRNA to structured regions and is beneficial in response to oxidative stress and DNA damage.

The manuscript is very well written and the line of thought is clearly laid out. The experimental approach is state of the art and, to the best of my understanding, well conducted.

Major criticism

1. The authors find that ProQ targets are protected from degradation when comparing RNA half-lives of wt and delta proQ mutants. They conclude from this finding that ProQ may counteract decay when binding to the 3'UTR of mRNAs. However, the presented data do not allow to assess if the RNA stabilising effect is really a specific or rather a secondary effect. As stated by the authors, also transcripts of cellular RNases are targets of ProQ (Pnpase) which makes it possible that the observed effects are global and not specific to ProQ binding. A relatively straightforward way to test this would be to select the ProQ binding site in 1-3 of the validation RNAs (*sodC*, *pnp*, *rpmG*, *AniS*,....) that was found in the CLIP-Seq peaks and replace it with a sequence that is not bound by ProQ. A repetition of the expression levels or half-lives after rifampicin treatment in wt or proQ ko (Fig. 4) would then allow to answer this question. This point must be addressed before suggesting ProQ having "a role in 3' end-dependent protection" (l.304).

Reply:

*We agree with the reviewer on the importance of supporting experimentally our prediction that ProQ protects against PNPase-mediated degradation of its targets. However, the in vivo experiments suggested by the reviewer are not as straight-forward as it may seem, firstly because of the difficult genetics of *N. meningitidis*, and secondly, from our previous experience with deleting ProQ sites in targets in *Salmonella* or *E. coli* transcripts which usually altered the decay pathway of those transcripts.*

*In order to prove more directly that ProQ can have a protective effect in the 3'UTR of targets, we have now cloned the *pnp* gene of *Neisseria* and used it to make recombinant PNPase to be able to perform an in vitro RNA degradation assay using ProQ and in vitro transcribed target RNA.*

*As shown in Supplementary Figure 12, PNPase degraded a 3' fragment of the *rpmG* mRNA efficiently in the absence of ProQ protein. By contrast, when this fragment was preincubated with ProQ, degradation was strongly impaired*

This establishes proof-of-concept for a ProQ-mediated protection against 3'->5' mediated RNA decay by PNPase. We describe the results of the in vitro protection assay in a short paragraph in the revised manuscript (l.249-254). In addition, we included a corresponding Material & Methods section termed "PNPase purification" and "RNA degradation assay" in the revised Supplement.

2. I am somewhat concerned about the experimental setup of the UV radiation experiment (comp. Fig. 6).

a) l.413/l.959: Why is the energy (20J) given in m^2 but the UV irradiation for the CLIP experiment in cm^2 ?

Reply:

We agree with the reviewer and changed the units from $20 J/m^2$ to $2 mJ/cm^2$ in order to be more consistent (l. 422 and Figure 6a in the revised manuscript).

b) 20J is a very high amount of energy, especially if one wants to squeeze that out of a Stratalinker as described. It must take several minutes to achieve such high irradiation dosages. How does this compare to the average turnover of the validation RNAs that were observed earlier? Is the treatment taking significantly longer than the effect on RNA you described earlier? In this respect, the details might matter a lot: was the negative control (0 J) treated in the same fashion (plated on a tray and left in the dark for the same time)? Please describe this in more detail in the Methods section.

Reply:

Actually, it takes 2.5 seconds to squeeze out $2 mJ/cm^2$ out of a Stratalinker. Therefore, the time of applied UV light treatment is much shorter than the average turnover of the validated RNAs, which exhibit RNA half-lives between 0.5 and 32 minutes. As suggested by the reviewer, we described the treatment of the negative control ($0 mJ/cm^2$) in the material and method part in more detail (l.422-424 in the revised manuscript).

c) l. 430: "Half of the culture was directly placed..." What happened to the other half?

Reply:

As now described in the revised material and methods section, half of the culture was directly placed in a 22×22 cm plastic tray and irradiated with UV light at $800 mJ/cm^2$ at 254 nm while the other half of bacterial cultures was stored in 50ml Greiner tubes on ice (l.441-442 in the revised manuscript).

Minor comments

3. l.117: Please add here that the chromosomal gene was "replaced/extended" with a version that has a FLAG tag (= no endogenous, non-tagged version of ProQ; not overexpressed from a plasmid). Its in the Methods section but stating it here more explicitly would make it much easier to read.

Reply:

According to the reviewer, we rephrased the section stating "Meningococci in which the chromosomal proQ gene was extended with a C-terminal 3xFLAG tag .." (l.118-119 in the revised manuscript) to make it easier to read.

4. l.252: You did not provide any experimental evidence that ProQ influences DNA damage repair in any fashion. Please provide such data or rephrase.

Reply:

We rephrased the sentence to "... and resistance to UV irradiation" (l.259 in the revised manuscript).

5. l.459: Please specify which type of SDS PAGE you used. Lämmli gels have been proven problematic for RNAs due to the high pH and using this system can lead to degradation of RNA.

Reply:

For SDS PAGE, we used the Tris-glycine pH 6.8/8.8 polyacrylamide gel system (Laemmli et al. 1970 Nature) as stated in the revised manuscript (l.469-470). While we appreciate the comment by this reviewer #1, we would like to emphasize that the protocol we are using has worked very well for other bacterial RBPs as well. Please see our previous work on the Salmonella CsrA, Hfq and ProQ proteins (Holmqvist E et al. 2016 EMBO J; Holmqvist et al. 2018 Molecular Cell), results of which were independently validate in E. coli by us and others. At this point in time, we have no indication that a possible effect of pH in the gel on RNA integrity would compromise the interpretation of our UV CLIP-seq results.

6. l.506: Typo: segemhel

Reply:

We thank the reviewer for identifying the typo and corrected it (l.516 in the revised manuscript).

7. l. 519: adjusted p-value. How was it adjusted?

Reply:

Benjamini-Hochberg corrected p-values as given now in the revised manuscript (l.528 in the revised manuscript).

8. Fig. 2a: I do not see the value in displaying the ProQ peaks over the genomic distribution. Is there any reason in Neisseria why it should bind to only distinct regions? Please clarify.

Reply:

Whereas Hfq has usually been identified as a global RNA-binding protein in distinct bacteria, the target suite of so far experimentally assessed FinO/ProQ-domain proteins is highly variable ranging from few RNA targets to being a global RNA binding protein. We investigated the targetome of the minimal meningococcal ProQ protein in order to understand the intrinsic targeting mode of the pure ProQ/ FinO domain.

Fig. 2a is intended to illustrate the point that the meningococcal ProQ is indeed a global RBP that targets dozens of RNAs from all over the bacterial chromosome and that therefore the ProQ/FinO domain is likely to bind rather specific than sensitive to RNA targets. We feel it is important to show this in order to make clear early on that the activity of N.m. ProQ is not restricted to a narrow region, e.g., a horizontally acquired part of the meningococcal genome.

9. In the same figure: why are the peaks shown for p 0.05 but the pie chart is using p 0.01?

Reply:

Using p 0.01 instead of 0.05 was a typo which we have now corrected (l. 857 and 860 in the revised manuscript).

10. In the same figure: The pie chart caption reads "RNA classes" but you show CDS/UTRs with sRNA and tRNA. Thats a bit confusing since the former are all mRNA features.

Reply:

We rephrased the title of panel b in Fig. 2 to mRNA feature and RNA class.

11. Fig. 3c: The y Axis is labeled "polyA" while I assume it should be "polyU". Also, I am not convinced that this data which is not statistically significant at all (l.179) is necessary to support your story.

Reply:

We thank the reviewer for identifying the typo and corrected it in panel e of Fig. 3. Although the differences are not statistically significant, there is a clear tendency for a shorter polyU tract length in ProQ-associated sRNAs. Because this is the first time a ProQ map has been reported for a species that is distinct from E. coli and Salmonella, we feel that this type of data is worth reporting.

12. Fig. 6: The labeling (a,b,c) of the panels and the text in the caption is all mixed up. Also, it is not clear if what happened to the 0 and 10J/m² data (they are not mentioned in the Methods part; see comment #2).

Reply:

We thank the reviewer for identifying the mixed up order of the figure legend and corrected it. As the 0 and 10 J/m² data are not included in the manuscript anymore, we deleted the phrase in the sentence.

Reviewer #2 (Remarks to the Author):

The manuscript by Bauriedl et al. concerns important topic of RNA recognition by the proteins with a ProQ/FinO domain, which are phylogenetically conserved in numerous β - and γ -proteobacteria bacteria, including human pathogens. Proteins of this family often have N- or C-terminal accessory domains besides the homologous ProQ/FinO domains. The RNA binding of proteins with a C-terminal extension (ProQ from E.coli and S.enterica, and RocC from L. pneumophila) or N- terminal extension (F-like plasmid FinO) have already been described. The manuscript by Bauriedl et al is the first that describes the RNA interactome of a protein composed solely of the ProQ/FinO domain. Comparing the RNA ligands of the single-domain N. meningitidis NMB1681 protein with those of two-domain ProQ or RocC provides the opportunity to elucidate what is the role of the ProQ/FinO domain in proteins from this family. The analysis of the structures of RNAs detected using the CLIP-seq study allowed the authors to conclude that the ProQ/FinO domain alone has a capability to recognize a subset of RNA molecules in the cell. This conclusion is important because it provides a new mechanistic insight into the function of the ProQ/FinO domain proteins, and it also allows to better understand the role of the ProQ homolog in Neisseria. Additionally, the authors characterized the physiological role of this protein, suggesting that its functions are related to those of Hfq, and indicating the physiological processes that are most affected by the ProQ homolog in Neisseria. Because of the general importance of the conclusions, this manuscript will be interesting to the wide readership of Nature Communications, and hence it is appropriate for the publication.

Specific comments

1) What is the likelihood of detecting RNAs that are not direct ligands of ProQ using CLIP-seq? For example, if an RNA was drawn into proximity of ProQ by pairing with another RNA, which is a specific

ligand, could it also be crosslinked and hence identified as a ligand using this method? Could this issue affect the outcome of the analysis?

Reply:

The unique advantage of UV CLIP is that it typically reports only zero-length induced covalent bonds between RNA and proteins. These bonds are introduced by UV crosslinking, which is extremely fast due to relaxation of the excited state in the picosecond time-frame (as discussed in following reviews (Budowski et al., 1986; Pashev et al., 1991; Shetlar, 1980). The combination of zero-distance and rapid crosslinking allows mostly for crosslinks between directly interacting partners and disallows multiple crosslinking between the RBP and other indirectly associated RNAs, in addition, collision events are even more unlikely to happen because the likelihood of interaction has to be multiplied with the likelihood of a crosslinking event. The efficiency of crosslinking is in general accepted to be below 1%, hence multiple crosslinking events on the same complex would yield a 0.01% probability. As RNA/cDNA sequencing offers an extreme dynamic range of quantification, these events could be possibly detected but were not considered in our data interpretation.

2) In Fig. 3e the plot is described as polyA distribution, instead of polyU distribution

Reply:

We thank the reviewer for identifying the typo and corrected it in panel e of Fig. 3.

3) The free RNAs in the binding assays on Fig. 4 are quite heterogenous. Could this correspond to alternative conformers?

Reply:

The RNAs were produced with T7 RNAP in vitro transcription, 5' labeling with ³²P-ATP, and extraction from a Urea-PAGE gel. Prior to the addition of ProQ, the RNAs are always boiled and cooled on ice that allows formation of secondary structures as are in general present in ProQ-targeted RNAs. We agree with the reviewer that additional higher bands of the sodC RNA most likely result from different conformations or oligomerization states. However, these represent a minor fraction of the RNA sample, any yet are equally shifted by ProQ in native-PAGE. In addition, yeast RNA is added to reduce background interactions that can also cause a heterogeneous population of sodC due to interaction.

Minor comments

Could you clarify the sentence on page 6, lines 168, 169?

Reply:

We rephrased the sentence: "Reciprocally, of the 401 Hfq-bound mRNAs (14) over a third code for proteins of unknown function." (l.171 in the revised manuscript)

In legend to figure 4a, did you mean "horizontal" instead of "vertical orange bar"?

Reply:

We thank the reviewer for identifying the typo and corrected it.

Mikołaj Olejniczak

Additional references:

Budowski, E., Axentyeva, M., Abdurashidova, G., Simukova, N., and Rubin, L. (1986). Induction of polynucleotide-protein cross-linkages by ultraviolet irradiation. *European Journal of Biochemistry* *159*, 95-101.

Pashev, I.G., Dimitrov, S.I., and Angelov, D. (1991). Crosslinking proteins to nucleic acids by ultraviolet laser irradiation. *Trends in Biochemical Sciences* *16*, 323-326.

Shetlar, M.D. (1980). Cross-Linking of Proteins to Nucleic Acids by Ultraviolet Light. In *Photochemical and Photobiological Reviews: Volume 5*, K.C. Smith, ed. (Boston, MA: Springer US), pp. 105-197.

REVIEWERS' COMMENTS:

Reviewer #1 (Remarks to the Author):

The authors have addressed all of my concerns to my satisfaction. Using a purified Pnpase to address my main criticism of RNA (de)stabilization via ProQ provides convincing additional results.

Reviewer #2 (Remarks to the Author):

The response of the Authors fully addresses my concerns. This is a thorough study, the results of which will be very useful for the bacterial RNA research community. This high-quality manuscript is certainly ready for publication.

Mikołaj Olejniczak